



# 1  The role of HONO in $O_3$ formation and insight into its formation

# 2  mechanism during the KORUS-AQ Campaign

Junsu Gil[1], Jeonghwan Kim[2], Meehye Lee[1], Gangwoong Lee[2], Dongsoo Lee[3], Jinsang Jung[4], Joonyeong
An[5], Jinkyu Hong[6], Seogju Cho[7], Jeonghoon Lee[8], Russell Long[9]
*[1] Department of Earth and Environmental Sciecne, Korea University, Seoul, Korea*
*[2] Department of Environmental Sciecne, Hankuk University of Foreign Studies, Yongin, Korea*
*[3] Department of Chemistry, Yonsei University, Seoul, Korea*
*[4] Korea Research Institute of Standards and Science (KRISS), Daejeon, Korea*
*[5] National Institute of Environmental Research (NIER), Incheon, Korea*
*[6] Department of Atmospheric Sciences, Yonsei University, Seoul, Korea*
*[7] Seoul Research Institute of Public Health and Environment, Seoul, Korea*
*[8] School of Mechanical Engineering, Korea University of Technology and Education, Cheonan, Korea*
*[9] United States Environmental Protection Agency, Washington, USA*
*Correspondence to*: Meehye Lee (meehye@korea.ac.kr)



**ABSTRACT**

Photolysis of nitrous acid (HONO) has long been recognized as an early morning source of OH radicals in
urban air, but the detailed mechanism of its formation is still unclear. During the Korea-US Air Quality (KORUS-AQ)
campaign, HONO was measured using Quantum Cascade Tunable Diode Laser Absorption Spectroscopy (QC-TDLAS)
at Olympic Park in Seoul from 17 May to 10 June, 2016. HONO concentrations ranged from 0.07 ppbv to 3.46 ppbv with
an average of 0.93 ppbv. HONO remained high at night from 1 am to 5 am, during which the mean concentration was
higher in high-$O_3$ episodes (1.82 ppbv) than non-episode (1.20 ppbv). In the morning, OH budget due to HONO photolysis
were higher by 50 % (0.95 pptv) during high-$O_3$ episodes compared to non-episode. Diurnal variations of $HO_x$ and $O_3$
simulated by the F0AM model demonstrated a difference of ~20 ppbv in daily maximum $O_3$ between the two periods.
The HONO concentration increased with relative humidity (RH) until 80 %, of which the highest HONO was associated
with the top 10 % $NO_x$, confirming that $NO_x$ is a crucial precursor of HONO and its formation is facilitated by humidity.
The conversion ratio of $NO_x$ to HONO was estimated to be $0.86\times10^{-2}$ $h^{-1}$ at night and also increased with RH. As surrogate
for the catalyst surface, the mass concentrations of black carbon (eBC) and the surface areas of particles smaller than 120
nm showed a tendency for RH similar to conversion ratio. Using an Artificial Neuron Network (ANN) model, HONO
concentrations were successfully simulated with measured variables (r = 0.8 for the best suite), among which $NO_x$, surface
area, and RH were found to be main factors affecting ambient HONO concentrations with weigh values of 26.2 %, 11.9 %,
and 10.6 %, respectively. This study demonstrates the coupling of HONO with $HO_x$-VOCs-$O_3$ cycle in Seoul Metropolitan
Areas (SMA) and provides practical evidence for heterogeneous formation of HONO by employing the ANN model to
atmospheric chemistry.




## 1 INTRODUCTION

The photolysis of nitrous acid (HONO) can severely impact the OH budget in the lower atmosphere (Brandenburger et al., 1998;Plass-Dülmer et al., 1998;Kotamarthi et al., 2001;Xing et al., 2019) through the following reaction.

$$HONO + h\nu \rightarrow NO + OH \ (300nm < \lambda < 405 \ nm) \tag{R1}$$

In recent studies, R1 was shown to contribute daily OH formation up to 30 %, and strongly impacted the early morning OH budget, which finally caused the oxidation capacity to increase, even in low HONO concentrations (Alicke et al., 2002;Ryan et al., 2018). HONO accelerates the morning oxidation process by transferring OH through the HONO-VOCs-$O_3$ chain and provides $NO_x$ (Alicke et al., 2003).This implies that there is a huge influence of HONO on the early morning photochemistry cycle, along with the promotion of VOC oxidation, causing high $O_3$ concentrations in afternoon (Aumont et al., 2003;Alicke et al., 2003;Kleffmann, 2007). Therefore, it is important to observe and predict the atmospheric HONO in order to understand the $HO_x$ ($HO_2$ + OH) and $NO_x$ (NO + $NO_2$) chemistry which influences $O_3$ production (Pitts and Pitts, 2000).

Although the importance of HONO to $O_3$ production was suggested in previous research, its role has not been fully understood due to the intricacies involved in HONO and $O_3$ formation under variety of ambient conditions. Recently, the high $O_3$ mechanism was a major concern in lots of studies. The Korea US – Air Quality (KORUS-AQ) campaign is one of these researches which made efforts to solve the photochemical pollution problem. KORUS-AQ campaign was conducted in May to June 2016, at Seoul Metropolitan Area (SMA) in South Korea. SMA is one of the most populated regions in the world, and it has been reported that SMA suffers from poor air quality caused by high concentrations of $O_3$ and $PM_{2.5}$ (Kim et al., 2018a;Kim et al., 2018b). In SMA, a better understanding of the chemical mechanisms is needed due to high $NO_x$-levels and dynamic change in meteorology which further complicates the photochemical process (Park, 2018;Kim et al., 2016;Ryu et al., 2013). During the KORUS-AQ campaign, airborne, satellite, and ground level measurements were conducted simultaneously within the same space-time frame. This provided a good opportunity for a comprehensive study of the chemical processes at high-$NO_x$ condition and overall insight into the photochemical cycle, encompassing gaseous and particle phase.

To measure the HONO, a number of techniques have been adopted. Annular Denuder (AD) coupled with Ion Chromatography (IC) was commonly used in the beginning (Ferm and Sjödin, 1985;Allegrini et al., 1987;Koutrakis et al., 1988;Appel et al., 1990;Komazaki et al., 1999). This collection system has been improved to Diffusion Denuder (DD) and Parallel Plate Diffusion Scrubber (PPDS), lowing detection limit to several ppt levels with less interference from other nitrogen species (Keuken et al., 1988;Simon et al., 1991;Simon and Dasgupta, 1993, 1995). This configuration has been utilized to date without any major changes (Takeuchi et al., 2004;Li et al., 2017c;Gu et al., 2009;Song et al., 2009;Kim et al., 2015). A Measuring AeRosols and GAses (MARGA) system is similar to PPDS-IC instruments at the point of measuring water-soluble trace gases as ionic species using a denuder. It can detect low concentrations and has shown clear variation, as have other comparison methods; however, it has been required to show improved accuracy until recently due to its artifacts (Makkonen et al., 2012;Stieger et al., 2018).

The introduction of spectroscopy technique has facilitated HONO measurement through instrumentation such as Chemical Ionization Mass Spectrometry (CIMS) (Fortner et al., 2004;Roberts et al., 2010;Levy et al., 2014), Cavity



Ring Down Spectrometry (CRDS) (O'Keefe and Deacon, 1988;Scherer et al., 1997;Wheeler et al., 1998), Differential
Optical Absorption Spectrometry (DOAS) (Winer and Biermann, 1994;Febo et al., 1996;Hendrick et al., 2014;Kleffmann
et al., 2006;Stutz et al., 2010;Garcia-Nieto et al., 2018;Perner and Platt, 1979), and Photo Fragmentation with Laser
Induced Fluorescence (PF-LIF) (Liao et al., 2006). All these methods have their own advantages and limitation. For
example, CRDS has the advantage of short time resolution, but has a relatively high detection limit (Wang and Zhang,
2000). PF-LIF has the strength to measure low concentrations and has a short time resolution, but shows high uncertainty
compared with other methods. Recently, Long Path Absorption Photometer (LOPAP) that is specialized to detecting high
HONO concentrations, has been employed for field measurements and chamber studies (Heland et al., 2001;Kleffmann
et al., 2006;Rohrer et al., 2005).
In addition, improved open-path spectroscopy, using Quantum Cascade Tunable Diode Laser Absorption
Spectroscopy (QC-TDLAS) has been applied in HONO measurements (Lee et al., 2011;Cui et al., 2018b). It is more
stable at room temperature without cooling, and easy to calibrate compared with using normal diode lasers. The inter-
comparison study of HONO instruments conducted in polluted condition demonstrated that there was general agreement
among all techniques and of these, TIDLAS instrument was used as a basis for pairwise comparison (Pinto et al., 2014).
The HONO formation mechanism is still not fully elucidated, albeit the progress in HONO measurement
techniques. There have been several suggestions for HONO formation reactions during the nighttime, but most models
have failed to accurately simulate HONO concentrations because various homogeneous and heterogeneous reactions are
involved in the production of HONO under certain conditions (Su et al., 2011;Sun et al., 2001;Aumont et al., 2003;Kim
et al., 2015;Su et al., 2008;Fu et al., 2019;Zhang et al., 2019a).

$NO + OH \rightarrow HONO$ (R2)
$NO_2 + H_2O \rightarrow HONO + OH$ (R3)
$NO_2 + ortho,para\text{-nitrophenols} \rightarrow HONO$ (R4)
$2NO_2 + H_2O + NH_3 \rightarrow HONO + NH_4NO_3$ (R5)
$2NO_2 + H_2O \rightarrow HONO + HNO_3$ (R6)
$NO_2 + NO + H_2O \rightarrow 2HONO$ (R7)
$NO + HNO_3 \rightarrow HONO + NO_2$ (R8)
$NO_2 + Soot \rightarrow HONO$ (R9)
$NO_2 + VOCs \rightarrow HONO$ (R10)

The reaction (R2) is well known as a HONO source in high NO and OH environments. This homogeneous
reaction is a major process for HONO production, but it is not sufficient to explain the HONO concentration in low OH
environments, especially during the nighttime (Kurtenbach et al., 2001). Therefore, other reactions (R3, R4, R5) were
suggested (Bejan et al., 2006;Li et al., 2008;Zhang and Tao, 2010;Barsotti et al., 2017), but the estimated HONO budget
still shows great uncertainty (Liu et al., 2019a). As a result, several studies suggested the possibility of HONO being
formed by heterogeneous reactions occurring on surfaces. Several laboratory and outdoor studies considered aqueous and
dry surfaces as reaction catalysts (Bari et al., 2003;Finlayson-Pitts et al., 2003;Wang et al., 2017;Spataro et al., 2017),
including aerosol (Hendrick et al., 2014;Tong et al., 2016;Bernard et al., 2016;Wang et al., 2016;Lu et al., 2018), black
carbon (Liang et al., 2017), and humic acid (Han et al., 2017;Yang et al., 2018) (R6~R10). Moreover, the surface includes



not only atmospheric particles but also ground soils, with the report of direct soil emission (Meusel et al., 2018;Bhattarai
et al., 2018). These heterogeneous conversions and emissions from soil are controlled by several conditions such as pH,
moisture, and microbials and mainly affect daytime HONO concentrations (Ermel et al., 2018;Wu et al., 2019). For these
reactions, the measurement of active surface area hinders the accurate estimation of HONO formation (Romer et al.,
2016). Recently, the photochemical reaction involving a nitrate series ($HNO_3$, $NO_3^-$) was considered as a major daytime
HONO source (Han et al., 2017;Li et al., 2018;Ye et al., 2017;Tsai et al., 2018;Cui et al., 2019). The HONO production
reaction is still controversial due to a wide variety of environments involving $NO_x$, RH, and surfaces and incomplete
understanding of multi-phase reaction (Bao et al., 2018;Zhang et al., 2019c;Wen et al., 2019;Zhang et al., 2019b). In this
context, new approaches are needed to elucidate the HONO formation mechanism.

In this study, we conducted a measurement and modeling at Olympic Park in Seoul for two purposes: To figure

out the photochemical processes responsible for high $O_3$, with an emphasis on HONO that contributes the early morning
OH budget, and to enhance the understanding of HONO formation mechanisms by evaluating the influence of key factors
on HONO variation. To achieve these objectives, we used a 0-dimensional photochemical model and newly introduced
the Artificial Neuron Network (ANN) method.

**2 METHODOLOGY**

**2.1 Measurement**

During the KORUS-AQ campaign, ground measurements were conducted at Olympic Park (37.57˚, 127.14˚)

in Seoul to get a comprehensive view of air quality (Figure 1). Olympic Park is 145 ha of natural green area located
southeast of the city center. The Han river flows northeast of the park, along which the Olympic Expressway extends. The
details about measurement will be found in Kim et al., 2019.

HONO was measured by the three institutions using the two techniques: Parallel Plate Diffusion Scrubber

coupled with Ion Chromatography (IC) system by Yonsei University and Korea University and Tunable infrared-laser
differential absorption spectrometer (TILDAS) with applied quantum cascade (QC) laser by Hankuk University for
Foreign Languages. All three measurements showed a reasonable correlation of r = 0.75~0.84. Because optical
measurement is free of sampling artifact with high time resolution (Pinto et al., 2014), HONO measurements by the QC-
TILDAS were used for further analysis in the present study.

QC-TILDAS was developed by Aerodyne Research Incorporation and is suitable for measuring highly reactive

trace gases, especially in the mid-infrared region, because of its high sensitivity, short response time (1 to 10 Hz), and
theoretically low detection limit (~0.1 ppb). It determines the mixing ratio of the target trace gas by monitoring its
molecular absorption at a certain wavenumber. In this study, we used 1276 $cm^{-1}$ for measuring HONO. HONO data were
collected every 1 seconds, and averaged hourly for assimilation with other measurement data. The absorption was then
compared with the theoretical spectrum of the HIgh-resolution TRANsmission (HITRAN) database to calculate the
mixing ratio of the trace gases. The Tunable Diode Laser Wintel data acquisition program (TDL Wintel) installed in the
instrument is designed to perform a frequency scan and acquire the resulting absorption spectrum, and then to analyze the
spectrum obtained by the instrument. Dry $N_2$ gas was injected every 5 minutes to clear out the Multi-Pass Cell (MPC)
and stabilize the baseline.

QC-TILDAS requires the management of physical conditions such as the pressure in the absorption chamber,





the temperature of the laser, and the temperature of the detector in order to keep the resolution spectrum stable. Large
noise levels were observed in the preliminary measurement data of QC-TILDAS probably due to the difficulty to manage
these physical conditions, especially due to the instability of the laser temperature. For this reason, the Kalman filter,
which is generally used for estimating and analyzing data from environment with large noise levels, was applied to
minimize the noise levels. First, assuming that the spectrum obtained by the non-negative least square method was the
reference spectrum, the Kalman filter was applied and then the HONO concentration was sequentially calculated by the
non-negative least square method again. All of these calculations were implemented in Python
Equivalent Black Carbon (eBC) was measured using Multi-Angle Absorption Photometer (MAAP, Thermo.
Inc) by Korea University of Technology and Education. Particle number concentration was measured using Scanning
Mobility Particle Sizer (SMPS, TSI. Inc) by National Institute of Environmental Research (NIER). Along with, were
measured reactive trace gases including $O_3$, $NO_x$, CO, VOCs, and HCHO, $PM_{2.5}$ mass, and meteorological parameters
including temperature, relative humidity (RH), and wind speed (Kim et al., 2019), and mixing layer height (MLH, (Lee
et al., 2019). Measurement periods started from 17[th] May to 10[th] June 2016, and all data were assimilated as 1 hour
averages.

**2.2 Model configuration**
**2.2.1 Framework for 0-D Atmospheric Modeling (F0AM)**

Framework for 0-D Atmospheric Modeling (F0AM) was developed by (Wolfe et al., 2016), which was
advanced version from the 1-D Chemistry of Atmosphere-Forest Exchange (CAFE). Its prior objective is to simulate the
chemical and physical process within the forest canopy, but it is also capable of tracing the change in chemical pollutants
in other environments. Due to the intrinsic nature of the model, it may overestimate the influence of the vegetation, but it
is suitable for this study because of the campaign period, which are most affected by the biogenic emissions. F0AM is
written in MATLAB and provides the option to choose the one of chemical reactions based on Master Chemical
Mechanism (MCM), Carbon Bond Mechanism (CB05), Regional Atmospheric Chemistry Mechanism (RACM), and
Goddard Earth Observing System – Chemical (GEOS-Chem) mechanism. In this study, we utilized MCMv3.3.1 with the
measured chemical and meteorological data sets which were hourly averaged. The dilution factor ($k_{dil}$) was adjusted and
a sensitivity test was conducted by excluding each factor. Finally, we quantified the impact of HONO on OH formation
and daily maximum $O_3$ concentration. The detailed conditions and results are discussed in session 3.3.

**2.2.2 Artificial Neural Network (ANN)**

The Artificial Neuron Network (ANN) was developed in computer science in the 1940s, but it has only been
recent years that successful applications could be possible because of the limitation of computational performance and
algorithm problems. Among various disciplines, ANN was integrated into atmospheric sciences to predict the movement
of airmass, meteorological change, and pollutant concentration variations (Shrivastava et al., 2012;Ong et al., 2016;Li et
al., 2017a;Li et al., 2017b;Nieto et al., 2018;Franceschi et al., 2018;Gardner and Dorling, 1998). ANN applied not only
in atmospheric science, but also in quantum chemistry for calculating energy state of HONO (Pradhan and Brown, 2017).
In comparison, there is little attempts to understand how the input data are related to output results in ANN, because it is
difficult to evaluate the weight of each neural network nodes. Therefore, in this study, we applied the ANN for the first



time to estimate the impact of each input factor to output HONO concentrations and approximately evaluate the weight
of each input species.

To easily construct the ANN model, we utilized powerful '*neuralnet*' packages in R (Riedmiller and Rprop,

1994;Anastasiadis et al., 2005). Description for ANN models are written in references, and we changed the stepmax from
$10^5$ to $10^8$ and repetitions from 1 to 3 for making calculation be possible and more accurate. Also, we fix the random
number sets using function offered from R for the repeatability of ANN model. To get the integrity of data, we selected
the data set from which all input factors are usable. Because the range of the measurement data set is largely different, all
data are normalized ($x_{nor}$) to the range of HONO using Eq. (2), application form of Eq. (1):

$x_{nor} = \frac{x - minimum(X)}{maximum(X) - minimum(X)}$                                                    (1)

$x_{nor}$ is the normalized value for $X$ using '*min-max scaling*' (Mohamad and Usman, 2013;Patel and Mehta,

2011). Then, $x_{nor}$ was adjusted to the HONO concentration scale following Eq. (2):

$x_{nor,HONO} = \frac{x - minimum(X)}{maximum(X) - minimum(X)} \times (maximum(HONO) - minimum(HONO)) + minimum(HONO)$    (2)

In a fully-connected artificial neural network (FC-ANN) which has only 1 hidden layer, the output (y) is

calculated by the Eq. (3) (Figure 2).

$y = \phi\left(h_{1,0}w_0^{out} + h_{1,1}w_1^{out} + h_{1,2}w_2^{out} + \cdots + h_{1,j}w_j^{out}\right) = \phi\left(\sum_{m=0}^{j} h_{1,m}w_m^{out}\right)$    (3)

The $h_{1,m}$ indicates the value of $m^{th}$ node in 1st hidden layer, and the $w_m^{out}$ indicates the weight of $m^{th}$ node

in a 1st hidden layer to the output. The terms expressed using 0 ($h_{1,0}$, and $w_0^{out}$) come from 'bias' terms, which were
represented as '+1' in the calculation. As similar as the propagation method from 1st hidden layer to output, each node
value in a 1st hidden layer can be shown as the result of the activation function which includes the sum of the multiple of
input layer variables ($x$) and weight ($w$) following Eq. (4):

$h_{1,m} = \phi\left(x_0 w_0^{h_{1,m}} + x_1 w_1^{h_{1,m}} + x_2 w_2^{h_{1,m}} + \cdots + x_i w_i^{h_{1,m}}\right) = \phi\left(\sum_{n=0}^{i} x_n w_n^{h_{1,m}}\right)$, where 1≤m≤j    (4)

In general, the results of the activation function are represented as $\phi(z)$, where z is the sum of multiple input

($x$, or $h$) and weights ($w$). Therefore, z can be written as $\sum_{m=0}^{j} h_m w_m^{out}$ or $\sum_{n=0}^{i} x_n w_n^{h_{1,m}}$, meaning that we can compare
the weight of each variable such as $w_m^{out}$ or $w_n^{h_{1,m}}$ because they are linearly coupled, and the result z is proportional to
the $\phi(z)$. However, there was a little issue for using the weight directly because the range and sign are different in each
weight. Therefore, we employed the softmax function, $softmax(w_p) = \frac{e^{w_p}}{\sum_{k=1}^{l} e^{w_k}}$, where $1 \le p \le l$. It is widely used for
measuring the portion of each variable p, due to its advantage that makes all variables in the range between 0 and 1. By
applying this function, we can compare the weight of each variable as Eq. (5) and Eq. (6):





weight of $h_{1,j}$ in hidden layer to y in output $(P_y^{h_{1,j}}) = \dfrac{e^{w_j^{out}}}{\Sigma_{m=0}^{j} e^{w_m^{out}}},$ (5)
weight of $x_i$ in input layer to $h_{1,j}$ in hidden layer $(P_{h_{1,j}}^{x_i}) = \dfrac{e^{w_i^{h_{1,j}}}}{\Sigma_{n=0}^{i} e^{w_n^{h_{1,j}}}},$ (6)

As a result, we can estimate the influence of $x_i$ to the final result (y) using Eq. (7):
weight of $x_i$ in input layer to y in output $= 100(\%) \times \Sigma_{m=1}^{j}(P_{h_m}^{x_i} \times P_y^{h_m}),$ (7)

**3 RESULTS AND DISCUSSION**

**3.1 Characteristic variation of HONO**

During the measurement, the average HONO concentration was 0.93 ppbv in the range of 0.07~3.46 ppbv, and

the average $O_3$ concentration was 40.6 ppbv in the range of 0.8~127.8 ppbv (Figure 3). When compared to the previous
measurement study, the HONO concentration in this study was lower than other urban sites (0.44~2.80 ppbv) (Table 1),
but it is obviously higher than suburban (0.28~0.66 ppbv) or rural (0.16~0.65 ppbv) sites.

In the entire experiment, there were 14 days when $O_3$ concentration exceeded 90 ppbv, close to the 95 %ile of

$O_3$ concentration (91.5 ppbv), which corresponds to the 'Unhealthy for sensitive groups' level of the Comprehensive Air-
quality Index (CAI). Thus, 17, 18, 19, 20, 22, 23, 25, 29, and 30 May, and 2, 5, 7, 9, and 10 June were categorized as
'high $O_3$ episodes', and the other 11 days were categorized as 'non-episodes'. $O_3$ and HONO concentrations were higher
in high $O_3$ episodes than non-episodes: average $O_3$ and HONO concentrations were 41.0 and 1.05 ppbv in high $O_3$ episodes,
and 40.1 and 0.81 ppbv in non-episodes, respectively. Especially, these differences were evident for the 95 %ile
concentration of $O_3$ and HONO, which were 94.7 and 2.58 ppbv during high $O_3$ episodes, and 79.7 and 1.91 ppbv for
non-episodes, respectively.

These High $O_3$ periods were distinguished by synoptic meteorological conditions: stagnant or transport period

in May and blocking period in June (Miyazaki et al., 2019). While the distribution of major chemical species was greatly
affected by the local or synoptic circulation of the atmosphere in May, the air mass was relatively homogeneous and aged
under domestic influence in June (Kim et al., 2019). It resulted in noticeable difference in chemical and meteorological
characteristics between May and June. For example, the daytime temperature was higher by 2.2 °C in May than June, and
the average CO concentration was higher by 178 ppbv in May than June. The $NO_x$ and VOCs concentrations were also
higher in May than June. The mean NO and $NO_2$ concentrations were 11.1and 30.1 ppbv in May and 6.9 and 25.6 ppbv
in June, respectively, leading to higher $NO_2/NO$ in June than May. The sum of Benzene, Toluene, Ethylbenzene, and o-
Xylene (BTEX) was 52.5 ppbC in May and 43.6 ppbC in June, on average. At lower precursor levels, however, the
daytime concentrations of HCHO and PAN were slightly higher in June than May. Likewise, the 95 %ile HONO
concentration at night was higher in June (2.41 ppbv) than May (2.39 ppbv). This result demonstrates that HONO was
intimately coupled with photochemical oxidation process during the KORUS-AQ campaign.

In general, HONO and $O_3$ showed an inverse correlation (Figure 4). In the present study, the overall correlation

between the two species was good ($r^2 = 0.41$). Interestingly, the nighttime HONO was higher during the high $O_3$ episodes
than other days. Comparing with the high $O_3$ episodes and non-episodes, average nighttime HONO concentrations (00~05
LST) was 1.82 ppbv in high $O_3$ episodes and 1.20 ppbv in non-episodes. BTEX concentration was higher at nighttime





than daytime on both high O₃ and non-episodes. In comparison, HCHO showed a clear peak in the morning and afternoon,
corresponding to the maximum concentration of $NO_x$ and $O_3$, respectively. It is likely that HONO was photolyzed in the
early morning, and the OH produced by HONO photolysis oxidized VOCs that was accumulated at night, generating
HCHO. Consequently, $O_3$ was formed as a final product, which will be discussed in terms of HONO-VOCs-$O_3$ chain in
section 3.2.
$NO_x$ showed a typical diurnal variation in urban areas with a rush hour peak at 8 am and started to increase
again at 4 pm shortly after $O_3$ reached its maximum. In particular, $NO_2$ remained high during the night, implicitly
indicating the possibility of $NO_2$-HONO interaction at high RH environment if high HONO was observed. While HONO
was positively correlated with $NO_x$, the correlation was better for $NO_2$ ($r^2 = 0.39$) than NO ($r^2 = 0.22$). HONO is also well
correlated with CO due to the common in diurnal variation with higher concentration at nighttime than daytime, in
accordance with the change in MLH. $PM_{2.5}$ and eBC showed an inversed pattern in their diurnal variation, albeit not clear.
It implies that there was contribution from secondary formation to $PM_{2.5}$, in addition to the influence of local emission,
airmass change, and transport.
HONO concentration increased with the increase of RH up to 80 %, where the maximum HONO was observed
(Figure 5). When RH was over 80 %, however, HONO concentration decreased. This type of RH dependency has been
reported in previous studies, and several hypotheses were suggested (Cui et al., 2018a;Huang et al., 2017;Li et al., 2012).
Figure 5 also illustrates that the highest HONO was associated with the top 10 % of $NO_x$ in all RH ranges except for those
with RH > 80 %. In Korea, RH = 80 % is the criteria that distinguishes haze and mist. Thus, it can be said that the
conversion of HONO from $NO_x$ was most efficient under haze conditions. Under high RH, the active surfaces available
for HONO formation would have been scavenged into mist particles. These results are convincing evidence for the active
role of $NO_x$ and aerosol surface in the formation of HONO. The detailed homogeneous-heterogeneous HONO formation
mechanism will be further discussed in section 3.3.

**3.2 $O_3$ formation through HONO-HO$_x$-VOCs mechanism under sunlight**

As we stated in a previous chapter, HONO concentration was higher during the high $O_3$ episodes than non-
episodes, and it was highlighted at nighttime (Figure 6). It is already known that HONO affects the daily OH budget, so
we compared the OH concentration produced from HONO photolysis (R1) between high $O_3$ episodes and non-episodes.
In previous studies, the steady-state HONO mixing ratio ([HONO]$_{ss}$) was calculated using the following Eq. (8) with
concentrations of NO ([NO]) and OH ([OH]), a photolysis rate constant of R1 ($J_{HONO}$), and reaction rate constant of R2
($k_2$) and R11 ($k_3$) (Kleffmann, 2007;Wong et al., 2012).

HONO + OH → NO + H₂O                                                                 (R11)
$[HONO]_{ss} = \frac{k_2[NO][OH]}{J_{HONO}+k_3[OH]},$                                         (8)

In this study, the OH mixing ratio was estimated using the measured HONO concentration, assuming that OH
is produced only from HONO. The photolysis rate constant of HONO ($J_{HONO}$) was calculated using Eq. (9), which was
developed by (Hayman, 1997), complemented by (Jenkin et al., 1997;Saunders et al., 2003), and incorporated in MCM
photolysis calculations (Wolfe et al., 2016).



$J = l\,(\cos(SZA))^m\,exp(-n\sin(SZA)),$        (9)

The constant $l$, $m$, and $n$ are 0.002644, 0.261, and -0.288, respectively. The calculated $J_{HONO}$ is in the range

between $0.6\times10^{-4}$ s$^{-1}$ and $0.2\times10^{-2}$ s$^{-1}$ with an average of $0.1\times10^{-2}$ s$^{-1}$, which is close to those of other measurement
references (Wong et al., 2012;Li et al., 2012). Finally, OH mixing ratio was calculated from Eq. (10):

$[OH] = J_{HONO}([HONO]_{t1} - [HONO]_{t2}),$        (10)

The estimated OH produced by HONO was significantly different between high $O_3$ episodes and non-episodes.

While the averaged OH concentration showed little difference in the afternoon (12:00~18:00 LST) between high $O_3$
episode (0.24±0.18 pptv) and non-episode (0.22±0.15 pptv), the OH concentration of the early morning (5:00~11:00 LST)
was noticeably higher in high $O_3$ episode (0.41±0.25 pptv) than non-episode (0.27±0.14 pptv). The integrated OH
concentration produced by HONO photolysis was also higher in high $O_3$ episodes ($\int_{early\ morning} OH\ dt$ = 2.87 pptv,
$\int_{afternoon} OH\ dt$ = 1.66 pptv) than non-episode ($\int_{early\ morning} OH\ dt$ = 1.92 pptv, $\int_{afternoon} OH\ dt$ = 1.56 pptv). This
simple estimation and comparison highlight the role of HONO in OH production.

In addition, F0AM was run with our measurements of HONO, BTEX, and HCHO for the period of June, to

quantitatively understand the role of HONO in HONO-HO$_x$-O$_3$ chain. In June, there were 5 days of high $O_3$ episodes and
details regarding measurements are stated in chapter 3.1. First, we adjusted the model configuration so that it properly
simulated the measured $O_3$ maximum and diurnal variation, which is a control run (S1) (Figure 7). Then, the model was
run with the three scenarios for comparison (Table 2). Without BTEX (S2), the maximum $O_3$ was decreased by 33.7 ppbv.
If HCHO as well as BTEX was not included (S3), the maximum $O_3$ concentration was lowered as large as 65.8 ppbv. The
S4 scenario without HONO reduced $O_3$ by 50.3 ppbv and shifted the maximum to morning. The result of sensitivity test
demonstrates the significant role of HONO in diurnal photochemical cycle.

For control scenario (S1), we estimated the contribution of HONO to HO$_x$ and $O_3$ concentration by comparing

high $O_3$ episodes with non-episodes in June (Figure 8). The nighttime HONO concentration was higher by 0.04~0.7 ppbv
in high $O_3$ episode than non-episode, promoting the production of HO$_x$ radicals: 0.1~0.2 pptv of OH in early morning and
late afternoon and 4.7~15.8 pptv of HO$_2$ in the late afternoon. Consequently, it resulted in increase in maximum $O_3$ by
about 20 ppbv. The results of F0AM model calculation confirm the role of HONO in HO$_x$ cycle in such that the photolysis
of HONO produces OH radical in the early morning, initiating the photochemical reaction involving VOCs and NO$_x$ and
facilitating the formation of $O_3$.

**3.3 Insight into HONO formation mechanism**

Despite the importance of HONO in photochemistry, detailed HONO formation mechanism is still not clear. It

has been proposed that NO, NO$_2$, and H$_2$O are intimately linked in HONO formation through physical mechanism as well
as chemical reactions. However, it is still difficult to quantitatively determine the contribution of each precursor to HONO
formation. Among all reactions, we examined the role of NO, NO$_2$ and RH in HONO formation through R3, R6, and R7.
Their relative importance was estimated simply by correlating HONO concentration with the product of all reactant





concentration as a surrogate of production rate. For all three reactions, the surrogate was positively correlated with HONO,
in which the low and high HONO was associated with low (~60 %) and high (60~90 %) RH, respectively. It suggests RH
is critical parameter determining HONO concentration under high $NO_x$ environment like Seoul.

Because HONO concentration stayed high at night from 0 am to 5 am (LST), the HONO formation mechanism

was investigated under this time zone, considering HONO photolysis, SZA, and MLH. In addition, we distinguished the
direct HONO emission ($[HONO]_{emission}$) from the chemical HONO formation ($[HONO]_{formation}$). The direct emission of
HONO from the tailpipe of vehicle was estimated as about 0.65 % of total $NO_x$ concentration (Kurtenbach et al., 2001;Liu
et al., 2017), which is commonly used in recent studies (Qin et al., 2009;Tong et al., 2015;Cui et al., 2018a). In the present
study, $[HONO]_{emission}$ is in the range from 0.02 to 0.53 ppbv with a mean of 0.17 ppbv. $[HONO]_{formation}$ is calculated by
subtracting $[HONO]_{emission}$ from measured HONO concentrations. Using this $[HONO]_{formation}$, we calculated the
conversion ratio of $NO_x$ ($C_{HONO,NO_x}$) to HONO from the following Eq. (11), which was application of suggested equation
in previous studies (Alicke et al., 2002;Alicke et al., 2003;Su et al., 2008;Hou et al., 2016).

$$C_{HONO,NO_x} = \frac{[HONO]_{cor,t2} - [HONO]_{cor,t1}}{(t_2 - t_1) \times [NO_x]}$$   (11)

It assumes that all HONO was only converted by NO and $NO_2$ in the nighttime. The average $NO_x$ to HONO

conversion ratio is $0.86 \times 10^{-2}$ $h^{-1}$. It is similar to the $NO_2$ conversion ratio of Shanghai (Cui et al., 2018a), and is in the
range of urban and rural areas which were reported in previous studies (Li et al., 2012;Xu et al., 2015;Huang et al.,
2017;Wang et al., 2015). As expected, the conversion ratio was increased along with the increase of RH until 80 %, as
$1.06 \times 10^{-2}$ $h^{-1}$ (Figure 9.a). However, the conversion ratio decreased when RH was over 80 %, like the HONO
concentration. This phenomenon already reported, and Wojtal et al., suggested that it was caused by the surface loss or
particle loss in marine boundary layers, which is in accordance with what we found in the present study discussed in
section 3.1 (Liu et al., 2019b;Wojtal et al., 2011).Therefore, the observed HONO is likely to be formed through
heterogeneous conversion which is mainly controlled by factors like $NO_x$, RH, and surface area.

In previous study, BC particles were suggested to serve as catalyst for heterogeneous reactions by providing

active sites for $H_2O$ and gaseous species owing to its complex microstructure (Zhang et al., 2008). The mass concentration
of eBC is available in this study and showed similar variation with $C_{HONO,NO_x}$ against RH (Figure 9.b). In urban areas,
the count median diameter (CMD) of fresh BC particle is typically found between 50 nm and 80 nm, and the range of
CMD is broadened approximately 30 to 120 nm when air mass is affected by a plume from aircraft engine exhaust or
wildfire (Petzold et al., 2005;Reddington et al., 2013). Based on number-size distributions obtained from the SMPS
measurement, the dry surface area was estimated, assuming that particles with a diameter of 30 nm to 121.9 nm likely
represent BC. Similarly to eBC and $C_{HONO,NO_x}$, the calculated surface area reached the maximum around RH = 80 %
(Figure 9.d). In comparison, $PM_{2.5}$ mass concentration was linearly increased with RH (Figure 9.c). In addition, there was
no consistent relationship between the total surface areas of particles smaller than 500 nm measured by SMPS and
$C_{HONO,NO_x}$, either.

Finally, we employed an Artificial Neural Network (ANN) method to test sensitivity and quantify the

contribution of $NO_x$, $H_2O$, and surface area on HONO formation (Figure 2). For model training and validating, we used
measurements of NO, $NO_2$, temperature (°C), RH (%), surface area of the 30~121 nm diameter range, wind speed, MLH,
and SZA.





To optimized node numbers in the hidden layer, we constructed the ANN from 8 to 20 of nodes which were in
the proper range of other references (Qiu et al., 2018). In addition, the k-fold cross validation method was applied for
more appropriate approach, using the number of k = 7. The performance of model was evaluated by the correlation
coefficient between the observed HONO (HONO$_{obs}$) and modeled HONO (HONO$_{mod}$). First, the measurement data was
divided into seven sets, of which the six subsets were used to train the ANN and the left one was to validate the result of
ANN. This process was repeated for all combinations of seven subsets with varying the number of nodes from 8 to 20
and iteration. Lastly, the correlation coefficient was averaged and the node number shown the highest coefficient was
selected. Consequently, the seven time of repeated training with 11 nodes resulted in the best correlation coefficient (r =
0.74). For the training sets with 11 nodes, the 6th iteration gave the highest correlation coefficient (r = 0.85).

In this configuration, ANN was employed to test the sensitivity of eight selected variables to HONO
concentration. As shown in Figure 10, the total weight of 8 variables was 61.1%. Of these, the weight of four variables
including NO, NO$_2$, RH, and surface area was substantial (> 10 %), in which the weight of NO$_x$ (26.2 %) was evidently
higher than that of surface area (11.9 %) or RH (10.6 %). The weight of other meteorology-related variables such as SZA
or MLH was relatively low, compare to those of chemical variables. It is probably because they are tightly coupled with
day-night cycle. The ANN model result is a convincing evidence for the heterogeneous formation of HONO from NO$_x$ in
urban atmosphere, albeit not fully explained by eight variables. There could be other sources than the three reactions
considered in this study such as soil emission or removal processes. Nonetheless, this study clearly demonstrates that the
ANN model realistically simulates the ambient HONO concentration using the measured variables and highlights that
NO$_x$, surface area, and RH are key factors for HONO formation in Seoul during the early summer.
**4 CONCLUSIONS**

To identify key mechanisms for high levels of O$_3$ and PM$_{2.5}$ in Seoul Metropolitan Areas (SMA) and to get solid
evidences for implementing policies, the KORUS-AQ (Korea-US Air Quality Study) campaign was conducted during
May ~ June 2016. As part of it, O$_3$ and trace gases including HONO were measured at Olympic Park in Seoul, a key
ground site.

HONO was measured using Quantum Cascade Tunable Diode Laser Absorption Spectroscopy (QC-TILDAS) at
1276 cm$^{-1}$ every 1 second, which was averaged for 1 h for subsequent analysis. The theoretical detection limit was better
than 0.1 ppbv. For the entire experiment, the HONO concentration ranged from 0.07 ppbv to 3.46 ppbv with the mean of
0.93 ppbv. HONO showed a typical diurnal cycle with the maximum at 4 am, and remained low but above the detection
limit during the day. The daily maximum concentration of HONO was different between the high-O$_3$ episodes and non-
episode. The high-O$_3$ episodes were selected for a total of 14 days based on the daily maximum O$_3$ of 90 ppbv (Kim et
al., 2019). The 95 %tile concentrations of O$_3$ and HONO were much higher in high O$_3$ episodes (94.7 ppbv and 2.58 ppbv)
than in non-episodes (79.1 ppbv and 1.91 ppbv). Similarly, the concentrations of NO$_x$, VOCs, and HCHO were higher in
high-O$_3$ periods than non-episodes. It implies that HONO is closely linked to the photochemical oxidation process that
produces O$_3$.

When OH concentration was calculated using $J_{HONO}$ assuming that OH production from HONO photolysis is
the only OH source, it was about 50 % higher in the early morning (5~11 am) during high-O$_3$ episodes (2.87 pptv)
compared to non-episodes (1.92 pptv). In addition, the photochemical F0AM model was utilized to simulate diurnal
photochemical cycle of HO$_x$ radicals and O$_3$ using the measurements of HONO and VOCs in June. The nighttime HONO



concentration of 0.3 ppbv that is the difference between the high-$O_3$ episodes and non-episode lead to the enhancement
of OH and $HO_2$ concentrations in the morning and afternoon, respectively, thereby increasing the maximum $O_3$
concentration by 20 ppbv.
The HONO concentration was increased with RH until 80 %, which was evident under high $NO_x$ condition (top
10 %). It implies that $NO_x$ is converted to HONO and this process is facilitated by humidity. The conversion ratio of $NO_x$
to HONO was estimated from the measured $NO_x$ and HONO concentrations and the average ratio was $0.86 \times 10^{-2}$ $h^{-1}$ at
night from 0 am to 5 am. The conversion ratio was highest at RH of 70~80 %. Similarly, the surface area of particles
between 30 nm and 121.9 nm in diameter and the mass concentration of eBC showed a similar trend for RH, as a surrogate
for aerosol surfaces where HONO formation can be catalyzed. Based on this empirical evidence, HONO concentrations
were successfully simulated in an Artificial Neural Network model with the eight measurement variables. Of these, NO,
$NO_2$, RH, and surface area were found to be the main factors that have the greatest impact on ambient HONO
concentrations. For the best suite of the ANN calculation and the measurement (r = 0.85), the weight values of $NO_x$, RH,
and surface area comprised 26.2 %, 10.6 %, and 11.9 %, respectively. This ANN model results demonstrate the
heterogeneous formation of HONO in SMA under high $NO_x$ condition. Consequently, ANN approach can be a useful
alternative to conventional model for investigating formation mechanisms of atmospheric constituents, which are not
unequivocally understood.
**5 AUTHOR CONTRIBUTION**

All authors participated in ground measurements at Olympic Park in Seoul during the KORUS-AQ campaign.
J. Gil, J. Kim, M. Lee, G. Lee, D. Lee measured HONO with difference methods. In particular, TILDAS system was
run by J. Kim and G. Lee. J. An established the platform for ground measurement. J. Jung, S. Cho, J. Hong, J. Lee, and
R. Long made measurements of $NO_x$ and CO, VOCs, mixing layer height, eBC, and HCHO, respectively. J. Gil and M.
Lee were responsible for model simulations and writing the manuscript.
**6 ACKNOWLEDGEMENTS**

This research was conducted as part of KORUS-AQ project, and study was financially funded by the National
Research Foundation of Korea (2017R1A2B4012143). Specially thanks to National Aeronautics and Space
Administration and National Institute of Environmental Research for supporting the experiment.





**7 TABLES AND FIGURES**

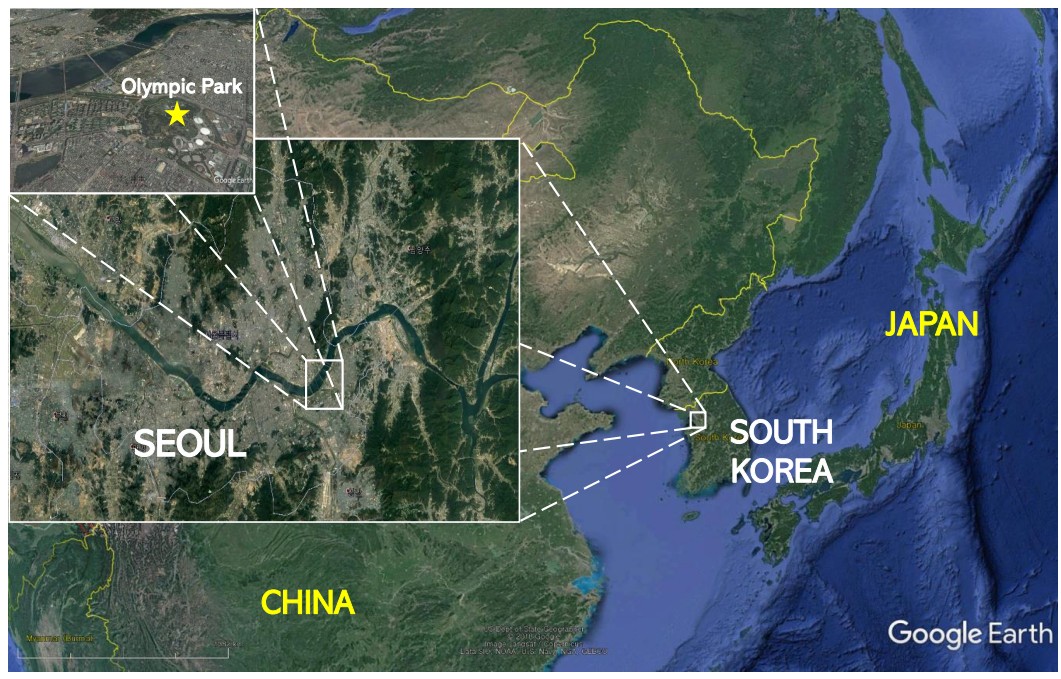


Figure 1. The map shows the location of Olympic Park as a key surface measurement site in Seoul, South Korea.
(© Google Earth)





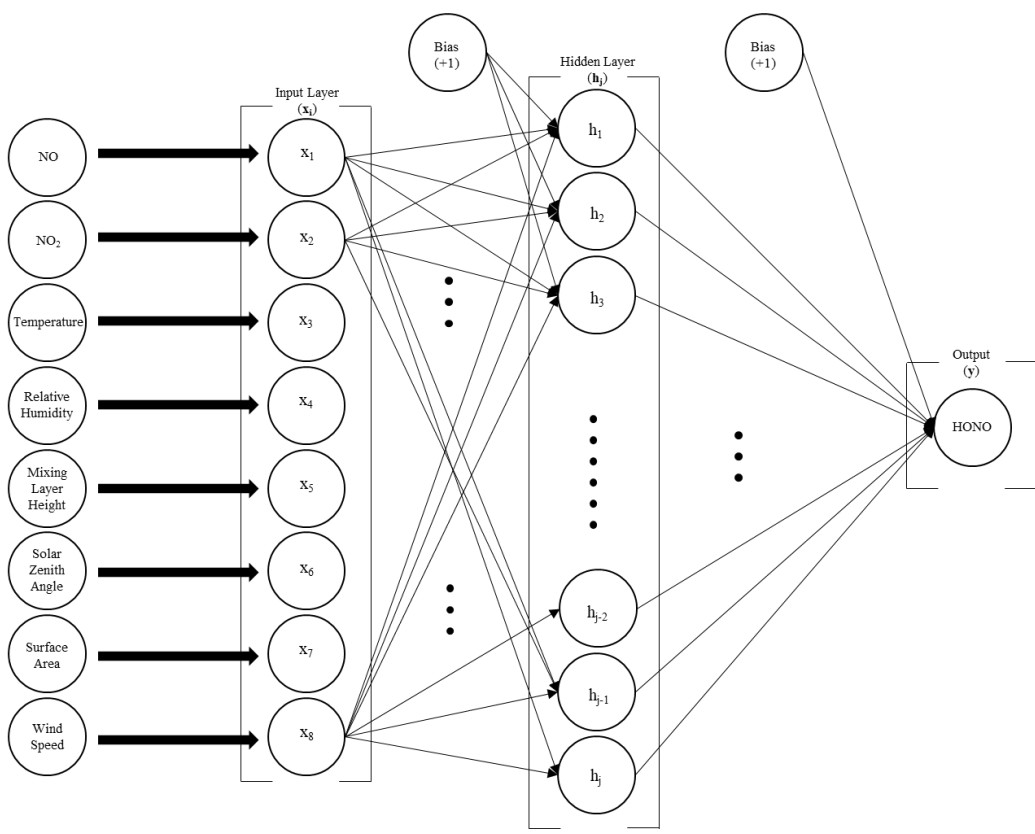


Figure 2. Structure of fully connected artificial neuron network (FC-ANN) model used in this study. In general ANN
expression, the equation of output is $y = \phi(\sum xw + b)$. In this study, we express the bias term '$b$' as $x_0 w_0$.



Figure 3. The variation of selected species measured at Olympic park during the KORUS-AQ campaign.





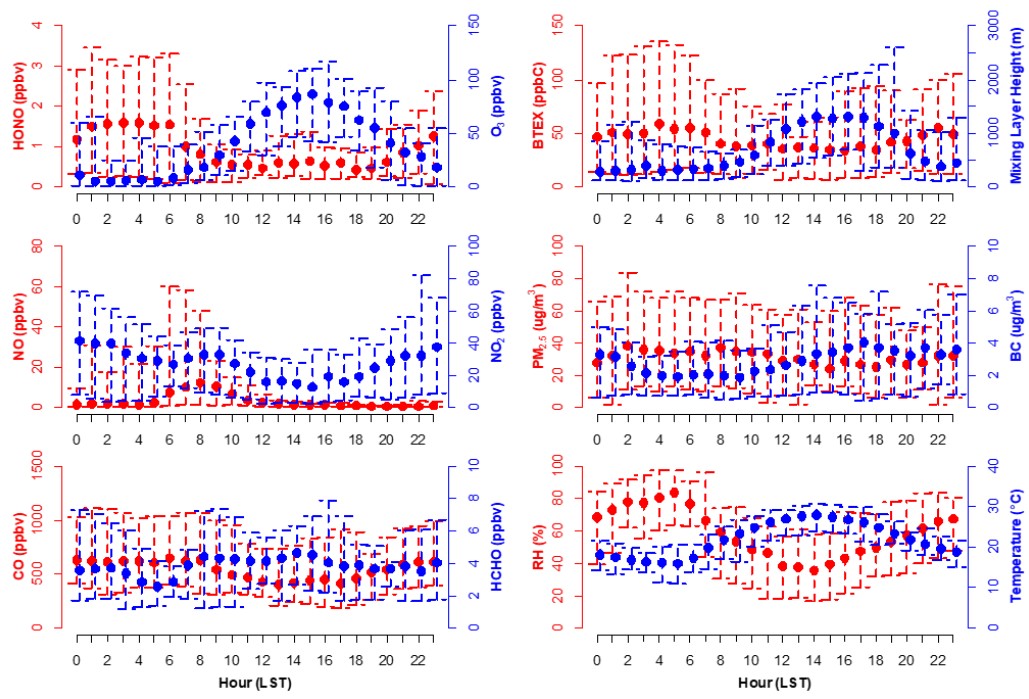


Figure 4. Diurnal variations of selected species measured at Olympic Park during the KORUS-AQ campaign.





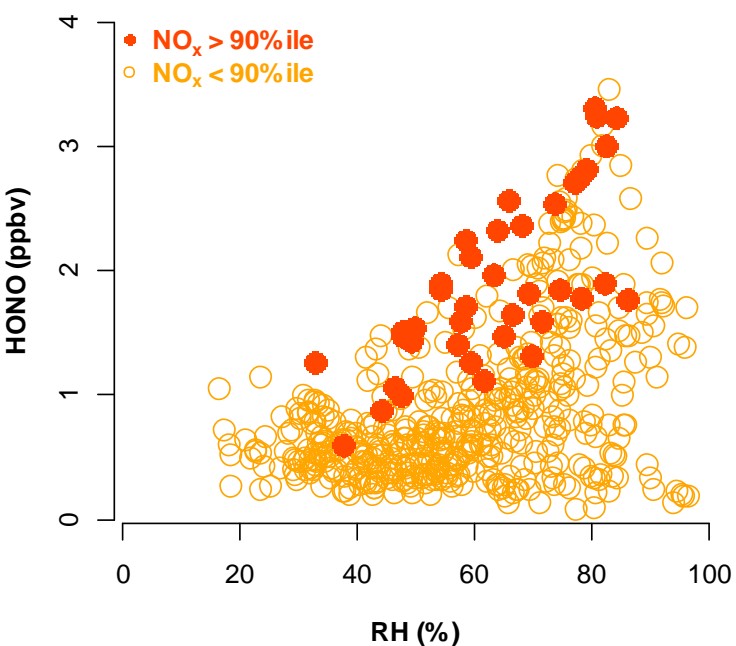


Figure 5. Correlation between HONO and RH color-coded with NO$_x$ level for entire measurements.

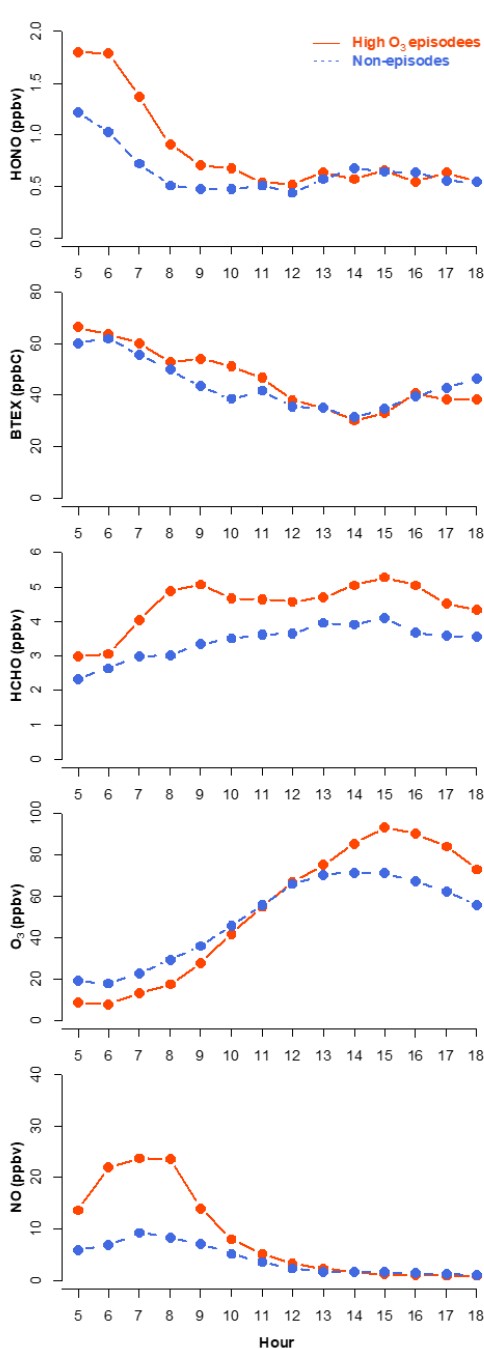


Figure 6. Diurnal variations of HONO, the sum of benzene, toluene, ethylbenzene, and xylene (BTEX), HCHO, $O_3$, and

NO during the high-$O_3$ episodes and non-episode for the entire experiment.





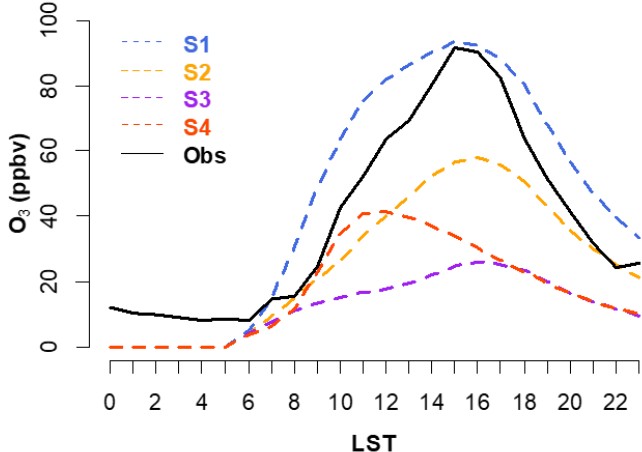


Figure 7. Diurnal variation of modeled $O_3$ in case of each scenario



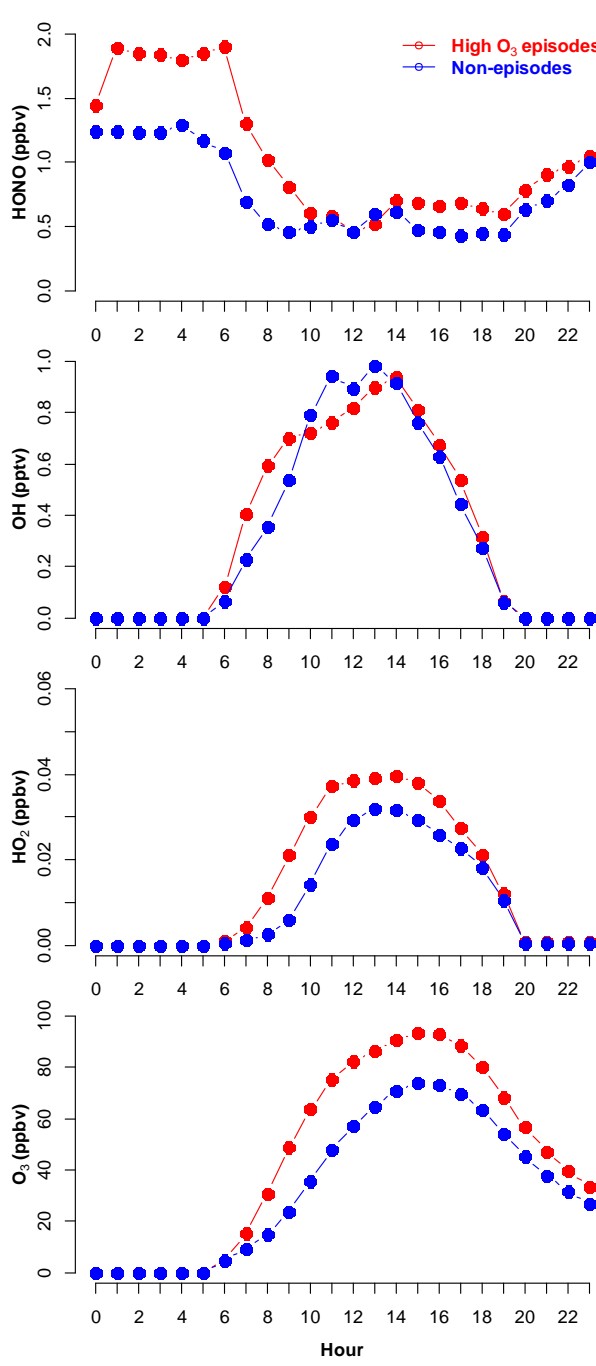


Figure 8: Diurnal variation of observed HONO, and modeled OH, HO₂, and O₃ using measurement data of high O₃ episodes and non-episodes





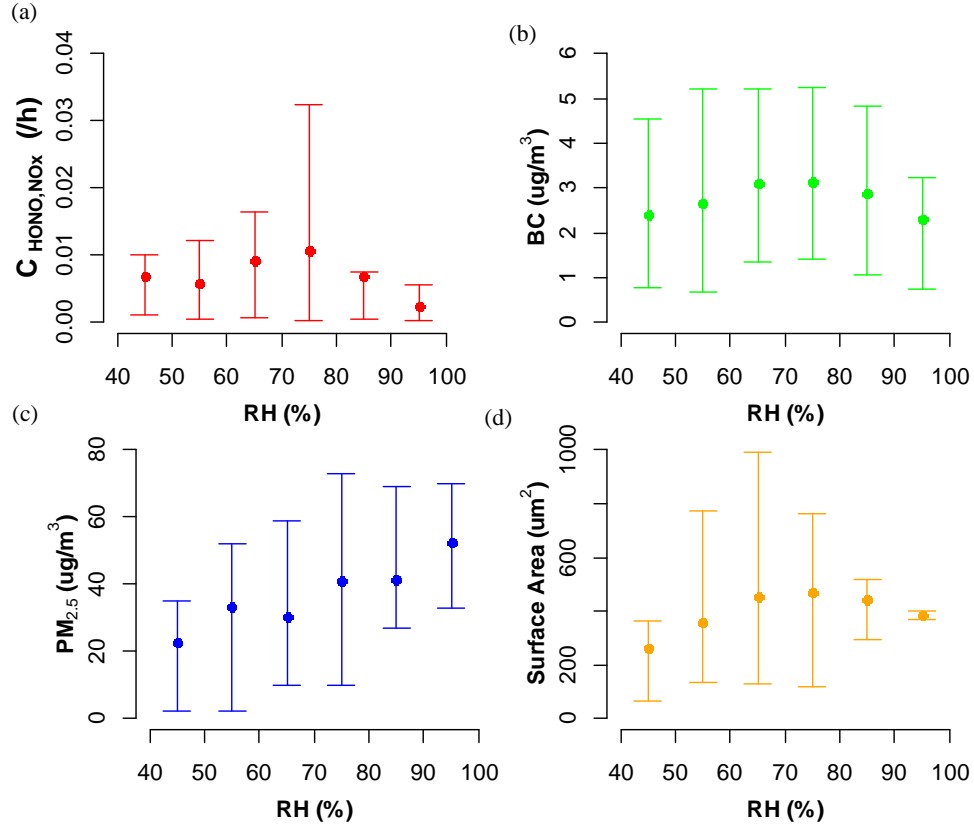


Figure 9. Relationship between (a) conversion ratio of NOx to HONO (see the text), (b) eBC mass concentration, (c)

PM$_{2.5}$ mass concentration, (d) surface area of particles between 30 to 120 nm diameter obtained from SMPS

measurement against relative humidity (RH). Solid circle represents the mean, and the upper and lower bar

stands for Q$_1$ - 1.5IQR, and Q$_3$ + 1.5IQR, respectively.





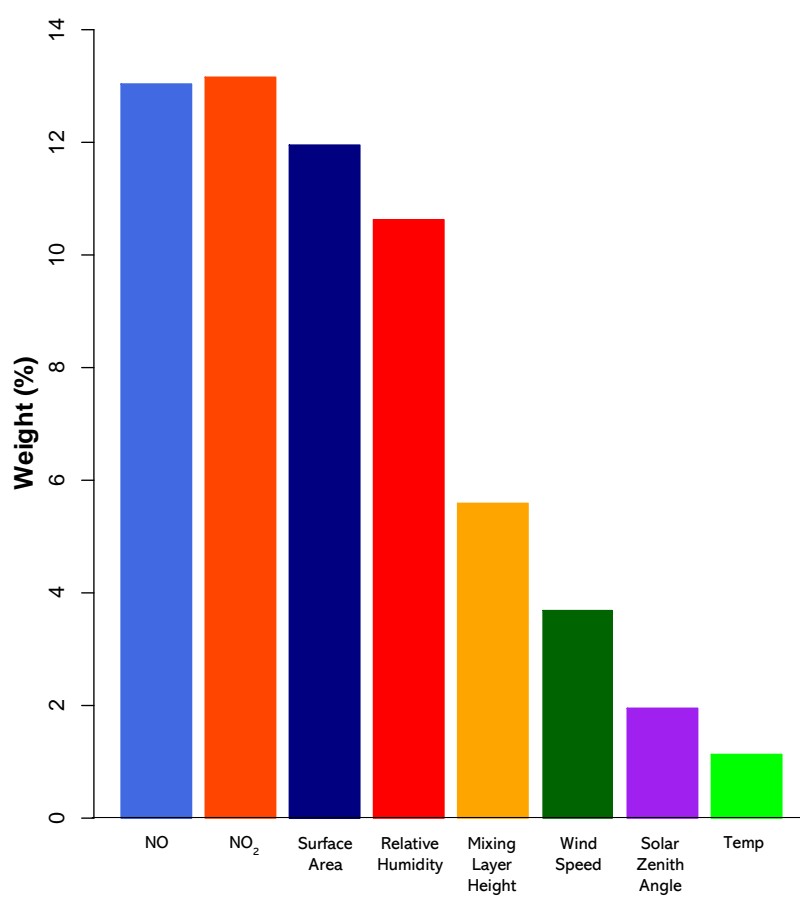


Figure 10. Weight percent of variables indicating their impact on HONO concentration in artificial neural network
(ANN) model.









Table 1. The summary of HONO measurements from previous studies.

| Type | Location | Period | Maen (ppbv) | Method* | Reference |
|------|----------|--------|-------------|---------|-----------|
| Urban | | | | | |
| | Hebei, China | Jun | 2.57 | SC-IC | (Xue et al., 2019) |
| | Shanghai, China | May | 2.31 | LOPAP | (Cui et al., 2018a) |
| | Kensington, UK | Jul~Aug | 1.05 | LOPAP | (Lee et al., 2016) |
| | Shanghai, China | Oct | 1.30 | LOPAP | (Bernard et al., 2016) |
| | Xi'an, China | Jul~Aug | 1.12 | LOPAP | (Huang et al., 2017) |
| | Guangzhou, China | Jul | 2.80 | DOAS | |
| | Beijing, China (Suburban~Urban) | Dec | 0.44~1.34 | DOAS/GAC | (Qin et al., 2009) |
| Suburban | | | | | |
| | Tungchung, HK | Summer | 0.66 | LOPAP | (Xu et al., 2015) |
| | Palaiseau, France | Jul | 0.28 | NitroMAC | (Michoud et al., 2014) |
| Rural | | | | | |
| | Wangdu, China | Jun~Jul | 0.49 | LOPAP | (Tan et al., 2018) |
| | Norfolk coast, UK | Jun~Jul | 0.16 | LOPAP | (Reed et al., 2016) |
| | Gyeong-Gi, South Korea | Jun | 0.65 | PPDS-IC | (Kim et al., 2015) |
| | Seoul, South Korea | Aug | 1.23 | LP-DOAS | (Lee et al., 2005) |
| | Seoul, South Korea | May~Jul | 0.36 | HEDS | (Song et al., 2009) |
| This study | | | | | |
| | | May~Jun | 0.93 | QC-TILDAS[#] | |

*Full name
# HONO was measured using PPDS-IC technique and their mean concentrations were lower than that of TILDAS.


Table 2. F0AM model configuration and four scenarios used in the present study.

| $k_{dil}$ : 21600 s$^{-1}$ (6 hr$^{-1}$) | S1 | S2 | S3 | S4 |
|------|------|------|------|------|
| HONO | O | O | O | X |
| BTEX (only o-xylene) | O | X | X | O |
| HCHO | O | O | X | O |
| *O$_{3mod}$/O$_{3obs}$ | 1.02 | 0.63 | 0.28 | 0.45 |

* Daily maximum concentration.



**8 REFERENCS**

Alicke, B., Platt, U., and Stutz, J.: Impact of nitrous acid photolysis on the total hydroxyl radical budget during the
Limitation of Oxidant Production/Pianura Padana Produzione di Ozono study in Milan, Journal of Geophysical Research:
Atmospheres, 107, LOP 9-1-LOP 9-17, 2002.
Alicke, B., Geyer, A., Hofzumahaus, A., Holland, F., Konrad, S., Pätz, H., Schäfer, J., Stutz, J., Volz-Thomas, A., and
Platt, U.: OH formation by HONO photolysis during the BERLIOZ experiment, Journal of Geophysical Research:
Atmospheres, 108, PHO 3-1-PHO 3-17, 2003.
Allegrini, I., De Santis, F., Di Palo, V., Febo, A., Perrino, C., Possanzini, M., and Liberti, A.: Annular denuder method for
sampling reactive gases and aerosols in the atmosphere, Science of the Total Environment, 67, 1-16, 1987.
Anastasiadis, A. D., Magoulas, G. D., and Vrahatis, M. N.: New globally convergent training scheme based on the resilient
propagation algorithm, Neurocomputing, 64, 253-270, 2005.
Appel, B., Winer, A., Tokiwa, Y., and Biermann, H.: Comparison of atmospheric nitrous acid measurements by annular
denuder and differential optical absorption systems, Atmospheric Environment. Part A. General Topics, 24, 611-616, 1990.
Aumont, B., Chervier, F., and Laval, S.: Contribution of HONO sources to the NOx/HOx/O3 chemistry in the polluted
boundary layer, Atmospheric Environment, 37, 487-498, 2003.
Bao, F., Li, M., Zhang, Y., Chen, C., and Zhao, J.: Photochemical Aging of Beijing Urban PM2. 5: HONO Production,
Environmental science & technology, 52, 6309-6316, 2018.
Bari, A., Ferraro, V., Wilson, L. R., Luttinger, D., and Husain, L.: Measurements of gaseous HONO, HNO3, SO2, HCl,
NH3, particulate sulfate and PM2. 5 in New York, NY, Atmospheric Environment, 37, 2825-2835, 2003.
Barsotti, F., Bartels-Rausch, T., De Laurentiis, E., Ammann, M., Brigante, M., Mailhot, G., Maurino, V., Minero, C., and
Vione, D.: Photochemical formation of nitrite and nitrous acid (HONO) upon irradiation of nitrophenols in aqueous
solution and in viscous secondary organic aerosol proxy, Environmental science & technology, 51, 7486-7495, 2017.
Bejan, I., El Aal, Y. A., Barnes, I., Benter, T., Bohn, B., Wiesen, P., and Kleffmann, J.: The photolysis of ortho-nitrophenols:
a new gas phase source of HONO, Physical Chemistry Chemical Physics, 8, 2028-2035, 2006.
Bernard, F., Cazaunau, M., Grosselin, B., Zhou, B., Zheng, J., Liang, P., Zhang, Y., Ye, X., Daële, V., and Mu, Y.:
Measurements of nitrous acid (HONO) in urban area of Shanghai, China, Environmental Science and Pollution Research,
541    23, 5818-5829, 2016.
Bhattarai, H. R., Virkajärvi, P., Yli-Pirilä, P., and Maljanen, M.: Emissions of atmospherically important nitrous acid
(HONO) gas from northern grassland soil increases in the presence of nitrite (NO2−), Agriculture, ecosystems &
environment, 256, 194-199, 2018.
Brandenburger, U., Brauers, T., Dorn, H.-P., Hausmann, M., and Ehhalt, D. H.: In-situ measurements of tropospheric
hydroxyl radicals by folded long-path laser absorption during the field campaign POPCORN, in: Atmospheric
Measurements during POPCORN—Characterisation of the Photochemistry over a Rural Area, Springer, 181-204, 1998.
Cui, L., Li, R., Zhang, Y., Meng, Y., Fu, H., and Chen, J.: An observational study of nitrous acid (HONO) in Shanghai,
China: The aerosol impact on HONO formation during the haze episodes, Science of The Total Environment, 630, 1057-
1070, 2018a.
Cui, L., Li, R., Fu, H., Li, Q., Zhang, L., George, C., and Chen, J.: Formation features of nitrous acid in the offshore area
of the East China Sea, Science of The Total Environment, 682, 138-150, 2019.
Cui, X., Yu, R., Chen, W., Zhang, Z., Pang, T., Sun, P., Xia, H., Wu, B., and Dong, F.: Development of a Quantum Cascade
Laser-based Sensor for Environmental HONO Monitoring in the mid-Infrared at 8 μm, Journal of Lightwave Technology,
2018b.
Ermel, M., Behrendt, T., Oswald, R., Derstroff, B., Wu, D., Hohlmann, S., Stönner, C., Pommerening-Röser, A., Könneke,
M., and Williams, J.: Hydroxylamine released by nitrifying microorganisms is a precursor for HONO emission from
drying soils, Scientific reports, 8, 1877, 2018.
Febo, A., Perrino, C., and Allegrini, I.: Measurement of nitrous acid in Milan, Italy, by DOAS and diffusion denuders,
Atmospheric Environment, 30, 3599-3609, 1996.
Ferm, M., and Sjödin, A.: A sodium carbonate coated denuder for determination of nitrous acid in the atmosphere,
Atmospheric Environment (1967), 19, 979-983, 1985.
Finlayson-Pitts, B., Wingen, L., Sumner, A., Syomin, D., and Ramazan, K.: The heterogeneous hydrolysis of NO 2 in
laboratory systems and in outdoor and indoor atmospheres: An integrated mechanism, Physical Chemistry Chemical
Physics, 5, 223-242, 2003.
Fortner, E. C., Zhao, J., and Zhang, R.: Development of ion drift-chemical ionization mass spectrometry, Analytical
chemistry, 76, 5436-5440, 2004.
Franceschi, F., Cobo, M., and Figueredo, M.: Discovering relationships and forecasting PM10 and PM2. 5 concentrations
in Bogotá, Colombia, using Artificial Neural Networks, Principal Component Analysis, and k-means clustering,
Atmospheric Pollution Research, 2018.
Fu, X., Wang, T., Zhang, L., Li, Q., Wang, Z., Xia, M., Yun, H., Wang, W., Yu, C., and Yue, D.: The significant contribution



of HONO to secondary pollutants during a severe winter pollution event in southern China, Atmospheric Chemistry and
Physics, 19, 1-14, 2019.
Garcia-Nieto, D., Benavent, N., and Saiz-Lopez, A.: Measurements of atmospheric HONO vertical distribution and
temporal evolution in Madrid (Spain) using the MAX-DOAS technique, Science of The Total Environment, 643, 957-
576    966, 2018.
Gardner, M. W., and Dorling, S.: Artificial neural networks (the multilayer perceptron)—a review of applications in the
atmospheric sciences, Atmospheric environment, 32, 2627-2636, 1998.
Gu, J., Zhang, Y., Zeng, L., and Wen, M.: Evaluation of the Performance of a Gas and Aerosol Collector (GAC), Research
of Environmental Sciences, 22, 16-22, 2009.
Han, C., Yang, W., Yang, H., and Xue, X.: Enhanced photochemical conversion of NO2 to HONO on humic acids in the
presence of benzophenone, Environmental pollution, 231, 979-986, 2017.
Hayman, G.: Effects of pollution control on UV exposure, AEA Technology Final Report prepared for the Department of
Health and Contract, 121, 6377, 1997.
Heland, J., Kleffmann, J., Kurtenbach, R., and Wiesen, P.: A new instrument to measure gaseous nitrous acid (HONO) in
the atmosphere, Environmental Science & Technology, 35, 3207-3212, 2001.
Hendrick, F., Müller, J.-F., Clémer, K., Wang, P., Mazière, M. D., Fayt, C., Gielen, C., Hermans, C., Ma, J., and Pinardi,
G.: Four years of ground-based MAX-DOAS observations of HONO and NO 2 in the Beijing area, Atmospheric
Chemistry and Physics, 14, 765-781, 2014.
Hou, S., Tong, S., Ge, M., and An, J.: Comparison of atmospheric nitrous acid during severe haze and clean periods in
Beijing, China, Atmospheric Environment, 124, 199-206, 2016.
Huang, R.-J., Yang, L., Cao, J., Wang, Q., Tie, X., Ho, K.-F., Shen, Z., Zhang, R., Li, G., and Zhu, C.: Concentration and
sources of atmospheric nitrous acid (HONO) at an urban site in Western China, Science of The Total Environment, 593,
594    165-172, 2017.
Jenkin, M. E., Saunders, S. M., and Pilling, M. J.: The tropospheric degradation of volatile organic compounds: a protocol
for mechanism development, Atmospheric Environment, 31, 81-104, 1997.
Keuken, M. P., Schoonebeek, C. A., van Wensveen-Louter, A., and Slanina, J.: Simultaneous sampling of NH3, HNO3,
HCl, SO2 and H2O2 in ambient air by a wet annular denuder system, Atmospheric Environment (1967), 22, 2541-2548,
599    1988.
Kim, H., Choi, W.-C., Rhee, H.-J., Suh, I., Lee, M., Blake, D. R., Kim, S., Jung, J., Lee, G., and Kim, D.-S.:
Meteorological and Chemical Factors Controlling Ozone Formation in Seoul during MAPS-Seoul 2015, Aerosol and Air
Quality Research, 18, 2274-2286, 2018a.
Kim, S., Kim, S.-Y., Lee, M., Shim, H., Wolfe, G., Guenther, A. B., He, A., Hong, Y., and Han, J.: Impact of isoprene and
HONO chemistry on ozone and OVOC formation in a semirural South Korean forest, Atmospheric Chemistry and Physics,
605    15, 4357-4371, 2015.
Kim, S., Sanchez, D., Wang, M., Seco, R., Jeong, D., Hughes, S., Barletta, B., Blake, D. R., Jung, J., and Kim, D.: OH
reactivity in urban and suburban regions in Seoul, South Korea–an East Asian megacity in a rapid transition, Faraday
discussions, 189, 231-251, 2016.
Kim, S., Jeong, D., Sanchez, D., Wang, M., Seco, R., Blake, D., Meinardi, S., Barletta, B., Hughes, S., and Jung, J.: The
Controlling Factors of Photochemical Ozone Production in Seoul, South Korea, Aerosol and Air Quality Research, 18,
2253-2261, 2018b
Kim, H., Lee, M., Gil, J., Jung, J., Long, W. R., Lee, G., Kim, D., Cho, S., Ahn, J., Hong, J., Park, M.: Overview and
charactersitics of air quality in Seoul Metropolitan Area during the KORUS-AQ campaign, Elementa: Science of the
Anthropocene, *be submitted*, 2019.
Kleffmann, J., Lörzer, J., Wiesen, P., Kern, C., Trick, S., Volkamer, R., Rodenas, M., and Wirtz, K.: Intercomparison of
the DOAS and LOPAP techniques for the detection of nitrous acid (HONO), Atmospheric Environment, 40, 3640-3652,
617    2006.
Kleffmann, J.: Daytime sources of nitrous acid (HONO) in the atmospheric boundary layer, ChemPhysChem, 8, 1137-
619    1144, 2007.
Komazaki, Y., Shimizu, H., and Tanaka, S.: A new measurement method for nitrogen oxides in the air using an annular
diffusion scrubber coated with titanium dioxide, Atmospheric Environment, 33, 4363-4371, 1999.
Kotamarthi, V., Gaffney, J., Marley, N., and Doskey, P.: Heterogeneous NOx chemistry in the polluted PBL, Atmospheric
Environment, 35, 4489-4498, 2001.
Koutrakis, P., Wolfson, J. M., Slater, J. L., Brauer, M., Spengler, J. D., Stevens, R. K., and Stone, C. L.: Evaluation of an
annular denuder/filter pack system to collect acidic aerosols and gases, Environmental science & technology, 22, 1463-
626    1468, 1988.
Kurtenbach, R., Becker, K., Gomes, J., Kleffmann, J., Lörzer, J., Spittler, M., Wiesen, P., Ackermann, R., Geyer, A., and
Platt, U.: Investigations of emissions and heterogeneous formation of HONO in a road traffic tunnel, Atmospheric
Environment, 35, 3385-3394, 2001.
Lee, B. H., Wood, E. C., Zahniser, M. S., McManus, J. B., Nelson, D. D., Herndon, S. C., Santoni, G., Wofsy, S. C., and





Munger, J. W.: Simultaneous measurements of atmospheric HONO and NO 2 via absorption spectroscopy using tunable
mid-infrared continuous-wave quantum cascade lasers, Applied Physics B, 102, 417-423, 2011.
Lee, C., Kim, Y. J., Hong, S.-B., Lee, H., Jung, J., Choi, Y.-J., Park, J., Kim, K.-H., Lee, J.-H., and Chun, K.-J.:
Measurement of atmospheric formaldehyde and monoaromatic hydrocarbons using differential optical absorption
spectroscopy during winter and summer intensive periods in Seoul, Korea, Water, Air, and Soil Pollution, 166, 181-195,
636 2005.
Lee, J., Whalley, L., Heard, D., Stone, D., Dunmore, R., Hamilton, J., Young, D., Allan, J., Laufs, S., and Kleffmann, J.:
Detailed budget analysis of HONO in central London reveals a missing daytime source, Atmospheric Chemistry and
Physics, 16, 2747-2764, 2016.
Lee, J., Hong, J.-W., Lee, K., Hong, J., Velasco, E., Lim, Y. J., Lee, J. B., Nam, K., and Park, J.: Ceilometer Monitoring
of Boundary-Layer Height and Its Application in Evaluating the Dilution Effect on Air Pollution, Boundary-Layer
Meteorology, 1-21, 2019.
Levy, M., Zhang, R., Zheng, J., Zhang, A. L., Xu, W., Gomez-Hernandez, M., Wang, Y., and Olaguer, E.: Measurements
of nitrous acid (HONO) using ion drift-chemical ionization mass spectrometry during the 2009 SHARP field campaign,
Atmospheric environment, 94, 231-240, 2014.
Li, L., Duan, Z., Li, H., Zhu, C., Henkelman, G., Francisco, J. S., and Zeng, X. C.: Formation of HONO from the NH3-
promoted hydrolysis of NO2 dimers in the atmosphere, Proceedings of the National Academy of Sciences, 115, 7236-
648 7241, 2018.
Li, S., Matthews, J., and Sinha, A.: Atmospheric hydroxyl radical production from electronically excited NO2 and H2O,
Science, 319, 1657-1660, 2008.
Li, T., Shen, H., Yuan, Q., Zhang, X., and Zhang, L.: Estimating Ground-Level PM2. 5 by Fusing Satellite and Station
Observations: A Geo-Intelligent Deep Learning Approach, Geophysical Research Letters, 44, 2017a.
Li, X., Brauers, T., Häseler, R., Bohn, B., Fuchs, H., Hofzumahaus, A., Holland, F., Lou, S., Lu, K., and Rohrer, F.:
Exploring the atmospheric chemistry of nitrous acid (HONO) at a rural site in Southern China, Atmospheric Chemistry
and Physics, 12, 1497-1513, 2012.
Li, X., Peng, L., Yao, X., Cui, S., Hu, Y., You, C., and Chi, T.: Long short-term memory neural network for air pollutant
concentration predictions: Method development and evaluation, Environmental Pollution, 231, 997-1004, 2017b.
Li, Z., Liu, Y., Lin, Y., Gautam, S., Kuo, H.-C., Tsai, C.-J., Yeh, H., Huang, W., Li, S.-W., and Wu, G.-J.: Development of
an automated system (PPWD/PILS) for studying PM2. 5 water-soluble ions and precursor gases: Field measurements in
two cities, Taiwan, Aerosol Air Qual. Res, 17, 426-443, 2017c.
Liang, Y., Zha, Q., Wang, W., Cui, L., Lui, K. H., Ho, K. F., Wang, Z., Lee, S.-c., and Wang, T.: Revisiting nitrous acid
(HONO) emission from on-road vehicles: A tunnel study with a mixed fleet, Journal of the Air & Waste Management
Association, 67, 797-805, 2017.
Liao, W., Hecobian, A., Mastromarino, J., and Tan, D.: Development of a photo-fragmentation/laser-induced fluorescence
measurement of atmospheric nitrous acid, Atmospheric Environment, 40, 17-26, 2006.
Liu, Y., Lu, K., Ma, Y., Yang, X., Zhang, W., Wu, Y., Peng, J., Shuai, S., Hu, M., and Zhang, Y.: Direct emission of nitrous
acid (HONO) from gasoline cars in China determined by vehicle chassis dynamometer experiments, Atmospheric
Environment, 169, 89-96, 2017.
Liu, Y., Lu, K., Li, X., Dong, H., Tan, Z., Wang, H., Zou, Q., Wu, Y., Zeng, L., and Hu, M.: A Comprehensive Model Test
of the HONO Sources Constrained to Field Measurements at Rural North China Plain, Environmental science &
technology, 2019a.
Liu, Y., Nie, W., Xu, Z., Wang, T., Wang, R., Li, Y., Wang, L., Chi, X., and Ding, A.: Contributions of different sources to
nitrous acid (HONO) at the SORPES station in eastern China: results from one-year continuous observation, Atmos.
Chem. Phys. Discuss., 2019, 1-47, 10.5194/acp-2019-219, 2019b.
Lu, X., Wang, Y., Li, J., Shen, L., and Fung, J. C.: Evidence of heterogeneous HONO formation from aerosols and the
regional photochemical impact of this HONO source, Environmental Research Letters, 13, 114002, 2018.
Makkonen, U., Virkkula, A., Mäntykenttä, J., Hakola, H., Keronen, P., Vakkari, V., and Aalto, P.: Semi-continuous gas
and inorganic aerosol measurements at a Finnish urban site: comparisons with filters, nitrogen in aerosol and gas phases,
and aerosol acidity, Atmospheric Chemistry and Physics, 12, 5617-5631, 2012.
Meusel, H., Tamm, A., Kuhn, U., Wu, D., Leifke, A. L., Fiedler, S., Ruckteschler, N., Yordanova, P., Lang-Yona, N., and
Pöhlker, M.: Emission of nitrous acid from soil and biological soil crusts represents an important source of HONO in the
remote atmosphere in Cyprus, Atmospheric Chemistry and Physics, 18, 799-813, 2018.
Michoud, V., Colomb, A., Borbon, A., Miet, K., Beekmann, M., Camredon, M., Aumont, B., Perrier, S., Zapf, P., and
Siour, G.: Study of the unknown HONO daytime source at a European suburban site during the MEGAPOLI summer and
winter field campaigns, Atmospheric Chemistry and Physics, 14, 2805-2822, 2014.
Miyazaki, K., Sekiya, T., Fu, D., Bowman, K., Kulawik, S., Sudo, K., Walker, T., Kanaya, Y., Takigawa, M., and Ogochi,
K.: Balance of Emission and Dynamical Controls on Ozone During the Korea-United States Air Quality Campaign From
Multiconstituent Satellite Data Assimilation, Journal of Geophysical Research: Atmospheres, 124, 387-413, 2019.
Mohamad, I. B., and Usman, D.: Standardization and its effects on K-means clustering algorithm, Research Journal of





Applied Sciences, Engineering and Technology, 6, 3299-3303, 2013.
Nieto, P. G., Lasheras, F. S., García-Gonzalo, E., and de Cos Juez, F.: PM 10 concentration forecasting in the metropolitan
area of Oviedo (Northern Spain) using models based on SVM, MLP, VARMA and ARIMA: a case study, Science of the
Total Environment, 621, 753-761, 2018.
O'Keefe, A., and Deacon, D. A.: Cavity ring-down optical spectrometer for absorption measurements using pulsed laser
sources, Review of Scientific Instruments, 59, 2544-2551, 1988.
Ong, B. T., Sugiura, K., and Zettsu, K.: Dynamically pre-trained deep recurrent neural networks using environmental
monitoring data for predicting PM2. 5, Neural Computing and Applications, 27, 1553-1566, 2016.
Park, M.-S.: Overview of Meteorological Surface Variables and Boundary-Layer Structures in the Seoul Metropolitan
Area during the MAPS-Seoul Campaign, Aerosol and Air Quality Research, 18, 2157-2172, 2018.
Patel, V. R., and Mehta, R. G.: Impact of outlier removal and normalization approach in modified k-means clustering
algorithm, International Journal of Computer Science Issues (IJCSI), 8, 331, 2011.
Perner, D., and Platt, U.: Detection of nitrous acid in the atmosphere by differential optical absorption, Geophysical
Research Letters, 6, 917-920, 1979.
Petzold, A., Gysel, M., Vancassel, X., Hitzenberger, R., Puxbaum, H., Vrochticky, S., Weingartner, E., Baltensperger, U.,
and Mirabel, P.: On the effects of organic matter and sulphur-containing compounds on the CCN activation of combustion
particles, Atmospheric Chemistry and Physics, 5, 3187-3203, 2005.
Pinto, J., Dibb, J., Lee, B., Rappenglück, B., Wood, E., Levy, M., Zhang, R. Y., Lefer, B., Ren, X. R., and Stutz, J.:
Intercomparison of field measurements of nitrous acid (HONO) during the SHARP campaign, Journal of Geophysical
Research: Atmospheres, 119, 5583-5601, 2014.
Pitts, B. F., and Pitts, J.: Chemistry of the upper and lower atmosphere: Theory, experiments and applications, Academic,
US, 2000.
Plass-Dülmer, C., Brauers, T., and Rudolph, J.: POPCORN: A field study of photochemistry in north-eastern Germany,
in: Atmospheric Measurements during POPCORN—Characterisation of the Photochemistry over a Rural Area, Springer,
714    5-31, 1998.

Pradhan, E., and Brown, A.: A ground state potential energy surface for HONO based on a neural network with
exponential fitting functions, Physical Chemistry Chemical Physics, 19, 22272-22281, 2017.
Qin, M., Xie, P., Su, H., Gu, J., Peng, F., Li, S., Zeng, L., Liu, J., Liu, W., and Zhang, Y.: An observational study of the
HONO–NO2 coupling at an urban site in Guangzhou City, South China, Atmospheric Environment, 43, 5731-5742, 2009.
Qiu, S., Chen, B., Wang, R., Zhu, Z., Wang, Y., and Qiu, X.: Atmospheric dispersion prediction and source estimation of
hazardous gas using artificial neural network, particle swarm optimization and expectation maximization, Atmospheric
Environment, 178, 158-163, 2018.
Reddington, C. L., McMeeking, G., Mann, G. W., Coe, H., Frontoso, M. G., Liu, D., Flynn, M., Spracklen, D. V., and
Carslaw, K. S.: The mass and number size distributions of black carbon aerosol over Europe, Atmospheric Chemistry and
Physics, 13, 4917-4939, 2013.
Reed, C., Brumby, C. A., Crilley, L. R., Kramer, L. J., Bloss, W. J., Seakins, P. W., Lee, J. D., and Carpenter, L. J.: HONO
measurement by differential photolysis, Atmospheric Measurement Techniques, 2483-2495, 2016.
Riedmiller, M., and Rprop, I.: Rprop-description and implementation details, 1994.
Roberts, J. M., Veres, P., Warneke, C., Neuman, J., Washenfelder, R., Brown, S., Baasandorj, M., Burkholder, J., Burling,
I., and Johnson, T. J.: Measurement of HONO, HNCO, and other inorganic acids by negative-ion proton-transfer
chemical-ionization mass spectrometry (NI-PT-CIMS): Application to biomass burning emissions, Atmospheric
Measurement Techniques, 3, 981, 2010.
Rohrer, F., Bohn, B., Brauers, T., Brüning, D., Johnen, F.-J., Wahner, A., and Kleffmann, J.: Characterisation of the
photolytic HONO-source in the atmosphere simulation chamber SAPHIR, Atmospheric Chemistry and Physics, 5, 2189-
734    2201, 2005.

Romer, P. S., Duffey, K. C., Wooldridge, P. J., Allen, H. M., Ayres, B. R., Brown, S. S., Brune, W. H., Crounse, J. D.,
Gouw, J. d., and Draper, D. C.: The lifetime of nitrogen oxides in an isoprene-dominated forest, Atmospheric Chemistry
and Physics, 16, 7623-7637, 2016.
Ryan, R. G., Rhodes, S., Tully, M., Wilson, S., Jones, N., Frieß, U., and Schofield, R.: Daytime HONO, NO 2 and aerosol
distributions from MAX-DOAS observations in Melbourne, Atmospheric Chemistry and Physics, 18, 13969-13985, 2018.
Ryu, Y.-H., Baik, J.-J., Kwak, K.-H., Kim, S., and Moon, N.: Impacts of urban land-surface forcing on ozone air quality
in the Seoul metropolitan area, Atmospheric Chemistry and Physics, 13, 2177-2194, 2013.
Saunders, S. M., Jenkin, M. E., Derwent, R., and Pilling, M.: Protocol for the development of the Master Chemical
Mechanism, MCM v3 (Part A): tropospheric degradation of non-aromatic volatile organic compounds, Atmospheric
Chemistry and Physics, 3, 161-180, 2003.
Scherer, J., Paul, J., O'keefe, A., and Saykally, R.: Cavity ringdown laser absorption spectroscopy: history, development,
and application to pulsed molecular beams, Chemical reviews, 97, 25-52, 1997.
Shrivastava, G., Karmakar, S., Kowar, M. K., and Guhathakurta, P.: Application of artificial neural networks in weather
forecasting: a comprehensive literature review, International Journal of Computer Applications, 51, 2012.



Simon, P. K., Dasgupta, P. K., and Vecera, Z.: Wet effluent denuder coupled liquid/ion chromatography systems,
Analytical Chemistry, 63, 1237-1242, 1991.
Simon, P. K., and Dasgupta, P. K.: Wet effluent denuder coupled liquid/ion chromatography systems: annular and parallel
plate denuders, Analytical Chemistry, 65, 1134-1139, 1993.
Simon, P. K., and Dasgupta, P. K.: Continuous automated measurement of gaseous nitrous and nitric acids and particulate
nitrite and nitrate, Environmental science & technology, 29, 1534-1541, 1995.
Song, C. H., Park, M. E., Lee, E. J., Lee, J. H., Lee, B. K., Lee, D. S., Kim, J., Han, J. S., Moon, K. J., and Kondo, Y.:
Possible particulate nitrite formation and its atmospheric implications inferred from the observations in Seoul, Korea,
Atmospheric Environment, 43, 2168-2173, 2009.
Spataro, F., Ianniello, A., Salvatori, R., Nardino, M., Esposito, G., and Montagnoli, M.: Sources of atmospheric nitrous
acid (HONO) in the European High Arctic, Rendiconti Lincei, 28, 25-33, 2017.
Stieger, B., Spindler, G., Fahlbusch, B., Müller, K., Grüner, A., Poulain, L., Thöni, L., Seitler, E., Wallasch, M., and
Herrmann, H.: Measurements of PM 10 ions and trace gases with the online system MARGA at the research station
Melpitz in Germany–A five-year study, Journal of Atmospheric Chemistry, 75, 33-70, 2018.
Stutz, J., Oh, H.-J., Whitlow, S. I., Anderson, C., Dibb, J. E., Flynn, J. H., Rappenglück, B., and Lefer, B.: Simultaneous
DOAS and mist-chamber IC measurements of HONO in Houston, TX, Atmospheric Environment, 44, 4090-4098, 2010.
Su, H., Cheng, Y. F., Cheng, P., Zhang, Y. H., Dong, S., Zeng, L. M., Wang, X., Slanina, J., Shao, M., and Wiedensohler,
A.: Observation of nighttime nitrous acid (HONO) formation at a non-urban site during PRIDE-PRD2004 in China,
Atmospheric Environment, 42, 6219-6232, 2008.
Su, H., Cheng, Y., Oswald, R., Behrendt, T., Trebs, I., Meixner, F. X., Andreae, M. O., Cheng, P., Zhang, Y., and Pöschl,
U.: Soil nitrite as a source of atmospheric HONO and OH radicals, Science, 1207687, 2011.
Sun, R., Hu, W., and Duan, Z.: Prediction of nitrogen solubility in pure water and aqueous NaCl solutions up to high
temperature, pressure, and ionic strength, Journal of solution chemistry, 30, 561-573, 2001.
Takeuchi, M., Li, J., Morris, K. J., and Dasgupta, P. K.: Membrane-based parallel plate denuder for the collection and
removal of soluble atmospheric gases, Analytical chemistry, 76, 1204-1210, 2004.
Tan, Z., Lu, K., Dong, H., Hu, M., Li, X., Liu, Y., Lu, S., Shao, M., Su, R., and Wang, H.: Explicit diagnosis of the local
ozone production rate and the ozone-NOx-VOC sensitivities, Science Bulletin, 63, 1067-1076, 2018.
Tong, S., Hou, S., Zhang, Y., Chu, B., Liu, Y., He, H., Zhao, P., and Ge, M.: Comparisons of measured nitrous acid (HONO)
concentrations in a pollution period at urban and suburban Beijing, in autumn of 2014, Science China Chemistry, 58,
778  1393-1402, 2015.
Tong, S., Hou, S., Zhang, Y., Chu, B., Liu, Y., He, H., Zhao, P., and Ge, M.: Exploring the nitrous acid (HONO) formation
mechanism in winter Beijing: direct emissions and heterogeneous production in urban and suburban areas, Faraday
discussions, 189, 213-230, 2016.
Tsai, C., Spolaor, M., Colosimo, S. F., Pikelnaya, O., Cheung, R., Williams, E., Gilman, J. B., Lerner, B. M., Zamora, R.
J., and Warneke, C.: Nitrous acid formation in a snow-free wintertime polluted rural area, Atmospheric Chemistry and
Physics, 18, 1977-1996, 2018.
Wang, G., Zhang, R., Gomez, M. E., Yang, L., Zamora, M. L., Hu, M., Lin, Y., Peng, J., Guo, S., and Meng, J.: Persistent
sulfate formation from London Fog to Chinese haze, Proceedings of the National Academy of Sciences, 113, 13630-
787  13635, 2016.
Wang, J., Zhang, X., Guo, J., Wang, Z., and Zhang, M.: Observation of nitrous acid (HONO) in Beijing, China: Seasonal
variation, nocturnal formation and daytime budget, Science of the Total Environment, 587, 350-359, 2017.
Wang, L., and Zhang, J.: Detection of nitrous acid by cavity ring-down spectroscopy, Environmental science & technology,
791  34, 4221-4227, 2000.
Wang, L., Wen, L., Xu, C., Chen, J., Wang, X., Yang, L., Wang, W., Yang, X., Sui, X., and Yao, L.: HONO and its potential
source particulate nitrite at an urban site in North China during the cold season, Science of the Total Environment, 538,
794  93-101, 2015.
Wen, L., Chen, T., Zheng, P., Wu, L., Wang, X., Mellouki, A., Xue, L., and Wang, W.: Nitrous acid in marine boundary
layer over eastern Bohai Sea, China: Characteristics, sources, and implications, Science of the Total Environment, 2019.
Wheeler, M. D., Newman, S. M., Orr-Ewing, A. J., and Ashfold, M. N.: Cavity ring-down spectroscopy, Journal of the
Chemical Society, Faraday Transactions, 94, 337-351, 1998.
Winer, A., and Biermann, H.: Long pathlength differential optical absorption spectroscopy (DOAS) measurements of
gaseous HONO, NO 2 and HCNO in the California South Coast Air Basin, Research on Chemical Intermediates, 20, 423-
801  445, 1994.
Wojtal, P., Halla, J., and McLaren, R.: Pseudo steady states of HONO measured in the nocturnal marine boundary layer:
a conceptual model for HONO formation on aqueous surfaces, Atmospheric Chemistry and Physics, 11, 3243-3261, 2011.
Wolfe, G. M., Marvin, M. R., Roberts, S. J., Travis, K. R., and Liao, J.: The Framework for 0-D Atmospheric Modeling
(F0AM) v3. 1, Geoscientific Model Development, 9, 3309, 2016.
Wong, K., Tsai, C., Lefer, B., Haman, C., Grossberg, N., Brune, W., Ren, X., Luke, W., and Stutz, J.: Daytime HONO
vertical gradients during SHARP 2009 in Houston, TX, Atmospheric Chemistry and Physics, 12, 635-652, 2012.



Wu, D., Horn, M. A., Behrendt, T., Müller, S., Li, J., Cole, J. A., Xie, B., Ju, X., Li, G., and Ermel, M.: Soil HONO
emissions at high moisture content are driven by microbial nitrate reduction to nitrite: tackling the HONO puzzle, The
ISME journal, 1, 2019.
Xing, L., Wu, J., Elser, M., Tong, S., Liu, S., Li, X., Liu, L., Cao, J., Zhou, J., and El-Haddad, I.: Wintertime secondary
organic aerosol formation in Beijing–Tianjin–Hebei (BTH): contributions of HONO sources and heterogeneous reactions,
Atmospheric Chemistry and Physics, 19, 2343-2359, 2019.
Xu, Z., Wang, T., Wu, J., Xue, L., Chan, J., Zha, Q., Zhou, S., Louie, P. K., and Luk, C. W.: Nitrous acid (HONO) in a
polluted subtropical atmosphere: Seasonal variability, direct vehicle emissions and heterogeneous production at ground
surface, Atmospheric environment, 106, 100-109, 2015.
Xue, C., Ye, C., Ma, Z., Liu, P., Zhang, Y., Zhang, C., Tang, K., Zhang, W., Zhao, X., and Wang, Y.: Development of
stripping coil-ion chromatograph method and intercomparison with CEAS and LOPAP to measure atmospheric HONO,
Science of the Total Environment, 646, 187-195, 2019.
Yang, W., Han, C., Yang, H., and Xue, X.: Significant HONO formation by the photolysis of nitrates in the presence of
humic acids, Environmental pollution, 243, 679-686, 2018.
Ye, C., Zhang, N., Gao, H., and Zhou, X.: Photolysis of Particulate Nitrate as a Source of HONO and NO x, Environmental
science & technology, 51, 6849-6856, 2017.
Zhang, B., and Tao, F.-M.: Direct homogeneous nucleation of NO2, H2O, and NH3 for the production of ammonium
nitrate particles and HONO gas, Chemical Physics Letters, 489, 143-147, 2010.
Zhang, J., An, J., Qu, Y., Liu, X., and Chen, Y.: Impacts of potential HONO sources on the concentrations of oxidants and
secondary organic aerosols in the Beijing-Tianjin-Hebei region of China, Science of the Total Environment, 647, 836-
852, 2019a.
Zhang, J., Chen, J., Xue, C., Chen, H., Zhang, Q., Liu, X., Mu, Y., Guo, Y., Wang, D., and Chen, Y.: Impacts of six
potential HONO sources on HOx budgets and SOA formation during a wintertime heavy haze period in the North China
Plain, Science of The Total Environment, 681, 110-123, 2019b.
Zhang, R., Khalizov, A. F., Pagels, J., Zhang, D., Xue, H., and McMurry, P. H.: Variability in morphology, hygroscopicity,
and optical properties of soot aerosols during atmospheric processing, Proceedings of the National Academy of Sciences,
834   105, 10291-10296, 2008.
Zhang, W., Tong, S., Ge, M., An, J., Shi, Z., Hou, S., Xia, K., Qu, Y., Zhang, H., and Chu, B.: Variations and sources of
nitrous acid (HONO) during a severe pollution episode in Beijing in winter 2016, Science of The Total Environment, 648,
253-262, 2019c.
