# Peer review of "The role of HONO in $O_3$ formation and insight into its formation"

_Atmospheric Chemistry and Physics, 2019_

## Referee Comment (RC1) · Anonymous Referee #1 · 11 Feb 2020

Nitrous acid (HONO) is an important precursor of OH radicals, which is closely related to the oxidation capability of the regional atmosphere, and thus promote a lot of atmospherically chemical process, such as secondary particle formation. In the manuscript, the authors performed a field-measured HONO in Korea, and proposed that the coupling of HONO with HOx -VOCs-O3 cycle in Seoul Metropolitan Areas (SMA), and suggested that NOx, surface area, and RH were the main factors affecting ambient HONO concentrations during field-observation in this region. The topic focused by this paper is very important in the field of atmosphere chemistry. However, the measurement period is so short, and it is difficult that the present data support the conclusion. The manuscript also suffered from a couple of big flaws. The whole manuscript was

not well organized and written. Especially, the data seems simple and was not well discussed. I thus refused this manuscript to be published on ACP.

1 In the INTRODUCTION section: Generally, the introduction section was poorly organized, and it should be rewritten. In this section, many literatures on the HONO field study were not cited, and the available studies were also not displayed in a line. Especially, the description about tools and methods should be positioned later on the HONO formation mechanism concluded from the field studies during the past years. In a word, this section should be written in more detail. 2 "In this study, we conducted a measurement... for two purposed:...1 to figure out ...2 to enhance ...", However, one can see that which did not coincide with the ABSTRACT structure. Why? 3 In the experimental section, ANN should also be mentioned in the INTRDUCTION sections, as well as the relevant literatures on HONO. 4 In the discussion section, Generally, the discussion about the field data is weak ( the measurement period is so short) and it is difficult to support the conclusion. 5 The English presentation is not so good, which could not be fulfil of the standard of the ACP manuscript. It should be improved greatly before publication. There are many spelling mistakes and syntactic error, as well as unsuitable sentence used. For example, in line 34. "...higher in high-$O_3$ episodes (1.82 ppbv) than non-episode (1.20 ppbv)" should be changed to "...higher in high-$O_3$ episodes (1.82 ppbv) than that in the non-episode (1.20 ppbv).

---

## Referee Comment (RC2) · Anonymous Referee #2 · 23 Feb 2020

The study describes measurements of HONO and associated atmospheric constituents in Seoul during KORUS-AQ. The authors find that elevated nocturnal HONO mixing ratios are detected on certain nights and that ground-level ozone mixing ratios are elevated during the following daytime period – a relationship that has been shown in the present study on several different days. The authors use a zero-dimensional box model to simulate this chemistry and show the effects of morning HONO photolysis on daytime O3 by jump-starting the HOx-NOx cycles with exogenous OH.

Overall, this paper leaves me thinking that I am unsurprised by the observation. The reason that HONO has received intense focus over the past number of years has been

to elucidate this type of mechanism – that elevated HONO in the morning would jump-start the HOx-NOx cycle, resulting in more rapid production of O3 during the day. In the end, I am still asking myself about what I have learned from this paper – and I am struggling to find very much. If the authors seek to provide a report of the atmospheric chemistry that is occurring in Seoul and contributing to high O3 days, this may be a worthy goal. I do not think that the present study provides new fundamental insight into HONO chemistry. It is recommended that the authors either more thoroughly utilize the data to extract useful quantitative information about HONO formation and provide insights that are more useful to the research community or back away from the HONO formation mechanism aspect of the study and focus more singularly on the impact that HONO can have on O3 chemistry in Seoul.

In its current state, this reviewer cannot recommend that this paper be published in Atmospheric Chemistry and Physics. If the paper were to be heavily revised (beyond what I would call a 'major revision'), the manuscript could be re-considered.

It is also recommended that the authors revisit their use of English language, perhaps hiring help to edit the manuscript if needed. Clearer language will only help the paper communicate the most important points.

Line 87: "The introduction of spectroscopy technique [sic] has facilitated measurement through instrumentation such as Chemical Ionization Mass Spectrometry (CIMS)..." CIMS is not an optical spectroscopy technique like the others listed. In addition, it is not discussed later in the paragraph when the positives and negatives are described.

Section 2.2.2 is difficult to understand. Parameters are weakly defined and technical terminology that is not typical for the atmospheric sciences makes the section challenging to understand for a likely ACP audience (including this reviewer). It is recommended that the authors include a description of how the choices made by the authors impact the outcomes of the ANN approach.

Lines 263-267: The average concentrations cited are not necessarily useful numbers

because the diel variations in mixing ratios are so large. The nocturnal HONO and daytime O3 mixing ratios are the most important, respectively. Do the 'average' values for the high-O3 and non-episode days reflect the day-long averages? If so, the authors should consider changing the presentation of data to be more applicable to the chemistry at hand.

Lines 280-282: The reason for the anti-correlation of O3 and HONO is due to the source and sink processes for these two gases. While the paper is suggesting that a relationship between these two important reactive trace gases exists, the sources of variation in the diel profile of HONO and O3 are actually *distantly* related. I do not understand why it is appropriate to discuss the data in such a simplistic way.

Figures 3 and 4: There is an important discrepancy between the HONO, NO, and NO2 data in Figs 3 and 4. It appears from inspection of many day-long periods in Figure 3 that NO and HONO are almost positively correlated. . . the peaks in NO and HONO occur at very similar times, but the data in Fig 4 show that these peaks occur at different times (NO is increasing as HONO is decreasing in the morning). Would it be better to plot the diel mixing ratios after splitting the cases into 'high-O3' and the 'non-episode' periods? The data, as plotted, are not necessarily consistent.

Line 301: Why does the 80% RH threshold only mean "mist" vs "haze" in Korea? This seems arbitrary. Perhaps consider the RH dependence of water uptake to urban aerosol particles (yes, a mixed and perhaps uncharacterized composition for sure, but there are typical particle types to consider qualitatively).

Figure 5: It may be clearer, or at least less arbitrary, to colorize the markers in Figure 5 by the NO mixing ratio. If a global relationship truly does exist, it will be borne out in the FULL dataset, rather than the arbitrarily chosen >90th percentile of NOx.

Line 309 and others: As presented throughout this paper, the HONO mixing ratios are said to be high during high-O3 episodes. Please alter these descriptions to be specific about how HONO is higher *at night* and is associated with elevated daytime ozone.

Eq 10: To what do the t1 and t2 subscripts refer?

Lines 319-320: "assuming that OH is produced only by HONO". Why do you make this assumption? The modeled diel OH curves shown cannot possibly be dependent only on HONO photolysis – what about the HOx-NOx cycle??

Section 3.2 up to Line 338: I think the problem here is in the way that this modeling experiment is presented. Please re-consider how this is presented and clearly indicate the question that you are asking, reasons for simplifying assumptions, and clear statements of conclusions based on this 'toy' experiment with modeling OH from your data. It appears that the authors are trying just to illustrate the impact of HONO on the morning OH mixing ratios, but their descriptions make it seem like their results are broadly accurate, which they are likely not based on their simplifying assumptions.

Lines 336-337: Time integrated values should have some indication of the limits of the integrals, otherwise these numbers lack real meaning. If the authors simply seek a comparison between the two types of events, then summarizing with a relative % difference may be more appropriate and useful.

Line 339-346: The sensitivity tests using F0AM say much more than the conclusion that the authors draw. The authors only comment on the importance of HONO to the O3 profile, but the other members of the HOx cycle also play extremely important roles – clearly shown by the model runs!! The authors are encouraged NOT to cherry-pick the results of their studies for the purported benefits of the story that they wish to tell. If the authors would like to highlight the importance of HONO, it would be beneficial to *quantify* the importance of HONO vs other factors.

Lines 357-364: The role of water in certain HONO production reactions is known. It is recommended that the authors explore RH vs absolute water vapor mixing ratio (or specific humidity). If water participates in the reaction, the kinetics will be related to the absolute concentration of water molecules, rather than a temperature-dependent, saturation-normalized metric like RH. Temperature changes by enough to drive diel

variations in RH. This may be masking the importance of water to the chemistry, or it may be over-emphasizing its role due to coincident diel changes in HONO, NO, NO2, and (temperature-driven) RH.

Lines 380-385: Two items: 1. The response of the conversion ratio to high RH is presented as an independent verification of the behavior of HONO under such conditions. The conversion ratio is essentially a ratio between a measure of [HONO] divided by a measure of [NOx]. If [NOx] is stable over time as it is in this study, then the [HONO]_emission metric is stable over time, and then of course the conversion ratio responds only to [HONO]_formation! The authors have simply shown two ways of expressing the same observation.

2. The connection to Wojtal et al is not very clear, because this is the first time that the marine boundary layer is described at all. Please be more thorough in the description of how the cited work helps to explain your observations.

Lines 386-396: Multiple comments: 1. Indeed, soot has been discussed as a reactive surface and laboratory experiments have shown that HONO can be formed upon NO2 reaction with soot. However, HONO can be formed by reaction with many different types of surfaces. Forcing the discussion into black carbon or soot particles reflects a narrow view of the surface catalyzed formation of HONO. In addition, making a broad assumption that all particles between 30-120 nm is FAR oversimplified and arbitrary. Particles in this size range will have a range of compositions, likely with a strong contribution from organic material. Why not simply cut the discussion of black carbon and use the total surface area of particles?

2. PM2.5 is discussed here but the measurement technique and sampling conditions are not described in the paper.

3. Guessing that PM2.5 is measured using a typical type of air quality monitor, PM2.5 could increase due to secondary aqueous phase reactions or simply the growth of particles due to water uptake. Also, particle mass increases as a cube of particle diameter, so PM2.5 measurements may be responding to an entirely different population of particles than the SMPE and MAAP. Also, if the sampling conditions for the SMPS and MAAP measurements are not identical to the PM2.5 measurements, the authors may be ascribing a real effect to a sampling artifact. Please provide sufficient technical information to reassure the reader that this is not the case. Particle measurements require care and nuance, which cannot be assessed by the reviewer or a future reader based on the present manuscript.

Lines 397-419: While the authors have used an interesting ANN tool, it is not clear what the results have taught us. We already know that the NOx, surface area, and humidity are key factors that control HONO production... we have known this for more than a decade. What can this information provide that will help advance the detailed understanding of HONO formation?

Minor overall comments: 1. The use of "%ile" is not typical in formal writing. The authors should consider changing this format from "95 %ile" to "95th percentile". 2. Line 309: "chapter" should be "section" (I stopped making these comments after this point. Substantial English language editing is advised.)

---

## Referee Comment (RC3) · Anonymous Referee #3 · 28 Feb 2020

General Comments

This paper presents an analysis of nighttime sources of nitrous acid (HONO) to the urban atmosphere of Seoul, Korea during the KORUS-AQ campaign through the use of an Artificial Neural Network (ANN) to model ambient concentrations. The manuscript has major flaws in the presented understanding of chemical mechanisms long established for importance to HONO nocturnal chemistry, and therefore, the analysis of the datasets and subsequent modeling are also compromised. The manuscript also suffers from unclear arguments and descriptions that makes many critical details almost impossible to follow for a valid review. This work requires major revisions to meet the

standard of publication in Atmospheric Chemistry and Physics, with particular attention paid to correct representation and analysis of nocturnal HONO chemistry, to which the presented dataset is limited. The revised manuscript will need to be further reviewed at that time.

Major Revisions

1. Nocturnal HONO formation chemical mechanisms are not correct. There is clear confusion in the presentation of what HONO production processes are relevant to exploring its nocturnal formation. The literature has not been sufficiently reviewed and puts the results of this study into question. In the introduction, the Authors present reactions 2 through 4 which have never been demonstrated in many prior reports to be important under nocturnal conditions. Further to this, reaction 4 is for the photolysis of ortho-nitrophenols, not from reaction of NO2. The Authors seem confused on this mechanism and it generates much skepticism regarding the quality of their model results if established HONO chemistry is presented erroneously here. Since release of HONO from ortho- and para-nitrophenols requires light, it is also invalid to explore for HONO production chemistry at night. This reaction has not been shown to be important in polluted daytime atmospheres for HONO production. How could it possible be important to nocturnal chemistry? Given the focus of this manuscript on nocturnal HONO production, the Authors need to revisit the state of knowledge for relevant nocturnal reactions to consider (e.g. has NO2 + NO + H2O ever made a significant amount of HONO in any prior report?) and rerun their ANN from a relevant starting point. Lines 119-123 in the Introduction are irrelevant to this work as this is all daytime HONO chemistry. Direct emissions from soils have not been quantified for diurnal roles (Lines 127-129) so are of limited utility. Seminal works for such investigations are also not cited correctly, a small selection includes: (Maljanen et al., 2013; Mushinski et al., 2019; Oswald et al., 2013; Scharko et al., 2015).

The Authors then proceed to describe heterogenous reactions as being universally catalytic, which simply is not true. Again, chemical mechanisms relevant to the daytime (photoexcited humic substances) are conflated with marginally relevant nocturnal sources (fresh soot emissions (Aubin and Abbatt, 2007)) raising major questions as to how the quality of the subsequent data analysis can be trusted. Further to this, there is a major oversight on at least 15 years of research reporting on the role of ground and aerosol surfaces in producing nocturnal HONO and the resultant vertical structure of HONO in the nocturnal atmosphere. A small selection of this body of research includes: (Kleffmann et al., 2003; Stutz et al., 2004; VandenBoer et al., 2013; Villena et al., 2011; Wong et al., 2011; Young et al., 2012). Nocturnal HONO production, in short, is a complex function of chemistry and mixing, often resulting in a complex vertical structure, which has shown in many polluted environments to be highly linked to ground surface conversion of NO2 with a water dependence. The Authors need to be clear in their understanding of the relevant chemical and physical processes governing nocturnal HONO production prior to exploring which is most important to O3 production via the ANN. Linking this understanding to the set up of the ANN is critical to validating the approach used in Section 2.2.2, for example. As this currently stands, this particular section is not possible to evaluate with a sound basis in HONO nocturnal chemistry and requires revision.

2. Given the focus of this work on the use of ANN to model nocturnal HONO chemistry, the Introduction and results should do a much better review and contrast to the existing capabilities of other models such as CMAQ (Czader et al., 2012; Diao et al., 2016; Gonçalves et al., 2012) or custom models (Kleffmann et al., 2003; Tsai et al., 2018; Vogel et al., 2003; Wong et al., 2012). The Authors should review this literature to aid in preparations for the re-running of their ANN.

3. The quality control for the HONO dataset, as presented in the methods section (and is entirely absent from the results/discussion) is not thorough. The Authors state that optical measurements are free of sampling artifacts, yet all instruments have inlets with surfaces where NO2 can react with water to produce HONO, a phenomenon described for most field instruments along with an applied correction for such an 'inlet effect'. The

best studies also account for non-linearities associated with changing NO2 and water concentrations. No sampling information for the HONO measurements is given. Were inlets used? How long were they? What were they made out of? What was the sample flow through the inlet? Was any particulate matter exclusion done on the sample flow? Was the NO2 inlet effect in producing HONO quantified and corrected for?

The intercomparison made here is also weak. An r-value of 0.74-0.84 is very bad and a correlation alone is insufficient to describe the comparability of the measurements. Slope, intercept, and their associated uncertainties are also necessary to calculate as part of measurement validation. These plots should be provided in a supporting information document. The Authors then go on to state that they made 1 Hz measurements of HONO and averaged these to 1 hour time points. Why was this done? Lines 168-172 suggest that the instrument was not operated correctly to make very good high time resolution measurements, yet no quality control analysis of the data is presented. How can one trust this HONO measurement? Justification for the approach used to arrive at a reliable final HONO dataset must be explained. Further to this, the Author's state a theoretical detection limit, but do not state what timescale this applies to. Presumably if they are averaging 1 Hz data to an hourly timescale, the detection limit is much better than 0.1 ppb? Or was the instrument operated so poorly that they were worse? Were field blanks performed on the instrument by overflowing the inlet with zero air? If yes, these should have been used to calculate the instrument detection limits that existed for this campaign, not the theoretical detection limits. Please perform these analyses to demonstrate that the HONO measurements are reliable and of high quality (or at least possible to clearly understand the potential limitations in their quality). As it currently stands, the Authors discuss data below the theoretical detection limit in Section 3.1 and one wonders how or why they would report this?

Detailed Comments

Line 30: 'but the…. Unclear'. This is not part of the manuscript objective. Remove from here.

Lines 32-46: Nearly the entire abstract has disjointed writing that is very difficult to follow the logic of. Revise according to major changes to findings/conclusions based on major revisions, but also for clarity. In particular, the sentence from Lines 39-41 makes no sense to me, even with a thorough understanding of HONO atmospheric chemistry. Also, at Lines 42-43 'surface area' is listed but what it applies to (i.e. aerosols or ground or both) is not clear.

Line 60: 'huge' is unnecessary to use here. Sensational modifiers throughout the manuscript should similarly be removed. The data will communicate the exact value and direction of changes, so Readers can evaluate this objectively.

Line 80: 'lowing' should be 'lowering'

Lines 83-86: If you are going to mention real-time IC techniques for HONO, there are earlier studies with better analytical results from the group of Jennifer Murphy (U Toronto) (VandenBoer et al., 2014). There is also a substantial amount of work from Jack Dibb (U New Hampshire) collecting HONO by mist chamber IC that have demonstrated good analytical capabilities in intercomparisons with DOAS (Pinto et al., 2014; Stutz et al., 2010). Also missing from this section is discussion of a substantial body of measurements made by broadband cavity enhanced absorption spectrometry (BBCEAS). Finally, given the lack of discussion on the performance and quality of the QC-TDLAS HONO measurements in this study, one is left wondering what the purpose of summarizing all of these analytical techniques is?

Lines 95-96: The Authors have the sensitivity of the LOPAP backwards. It is very good a detecting 'low mixing ratios of HONO'. The detection limits of the LOPAP are often below 1 pptv if integrated over 5 minutes or more. Furthermore, there are many custom-built stripping coil instrument references that are missing here (Ren et al., 2011; Ye et al., 2016; Zhang et al., 2009).

Line 102: 'TIDLAS' should be 'TDLAS'

[Figure]

Line 103: 'albeit' should be 'despite'

Lines 152-154: The TDLAS technique has been described three times up to this point in the manuscript and denoted with three different descriptions and abbreviations. Pick one and use it throughout.

Line 187: Reference formatting is incorrect and needs to be revised throughout.

Lines 195-197: Why is the dilution factor the only parameter adjusted in the model that is worth mentioning in the methods? This sensitivity test comes out of nowhere and no justification is provided for why this is a sound approach.

Line 197: 'session 3.3' should be 'Section 3.3'

Lines 199 – 201: The Authors define Artificial Neural Network (ANN) over and over. Do it once at the first mention and simply use ANN moving forward. The second instance of its use at Line 201 has been written incorrectly.

Lines 205-206: The use of ANN for quantum states of HONO is irrelevant to this work. Remove.

Line 208: Revise to '...of each neural network node with a clear chemical or physical process'

Line 211: Check citations for correctness of Author names and all other details according to ACP guidelines for Authors.

Lines 216-218: But what are you doing, and why, with the calculation in Equation 1? This is very unclear. Revise.

Line 260: '%ile' is incorrect. This is 'percentile'. Fix throughout.

Line 264: 'average O3 and HONO' What kind of average is this? 24-hour average? Clarify. The presented data does not look like it has any 24-hr period with more than 1 ppbv of HONO. Line 268: 'High' should be 'high'

Line 277: The dataset for HONO would be expected to be log-normal distributed. Here and above, the authors are discussing the 95th percentile with an implicit meaning of significance, yet if the dataset is not log-transformed, then the 95th percentile would be a biased value to report due to the skewed distribution from a normal distribution.

Lines 291-294: This is suggestive of strong direct emissions. See many recent papers from Beijing and perform the same analysis for Seoul (Liu et al., 2017; Tong et al., 2016; Wang et al., 2017) instead of choosing an estimate for this work.

Line 304: The Authors conclude here that aerosol surfaces are important, but their lack of awareness of literature reports on the role of the ground surface relative to aerosol surfaces being far more important suggests that they need to reconsider their interpretation of the data after more thoroughly reviewing the literature (see references given in Major Comments above). All vertically-resolved HONO measurements have shown that the ground is a major sink for HONO, as well as dew and available surface water (He et al., 2006; Rubio et al., 2009; Stutz et al., 2004; Tsai et al., 2018; VandenBoer et al., 2015; Wojtal et al., 2011).

Line 319-320: How can the Authors justify ignoring the photolysis of O3 and aldehydes as major sources of OH during KORUS-AQ. This does not seem reasonable. Even in winter atmospheres, this additional chemistry leading to OH production has been shown to be critical to capture (Kim et al., 2014).

Lines 336-337: Why is this integral notation being used? The discrete time intervals being used are not given and the explanation here is very unclear. Revise.

Lines 341-346: Add line colors to the S# references here so they do not get confused with figures in the supplemental information. Or change the identification codes for the model runs to not start with 'S'.

Lines 357-359: 'physical mechanisms as well as'. This is nonsense. Heterogeneous hydrolysis is a chemical mechanism. Remove.

Line 360: Exploring the role of NO (and R7), based on all known nocturnal HONO chemistry is useless and should be removed throughout the manuscript. There is no basis for this from the literature.

Line 366: 'considered HONO photolysis, SZA, and MLH' This is nighttime data. Why is photochemistry being considered? The reasoning here is almost impossible to understand.

Lines 371-373: It is the conversion of NO2 that is typically used for this calculation, not NOx, which is consistent with the hydrolysis of NO2 being the dominant chemical source of HONO to the nocturnal atmosphere. The Authors should use NO2 conversion efficiency in this work. Given the discussion that follows on Lines 377-380 about NO2 conversion, perhaps this is a typo?

Line 381: Figure 9a. Fix notation.

Lines 382-385: The work cited here made measurements over open marine water. The dataset in this work is collected over an urban landscape. The uptake of NO2 to ground surfaces has been described in detail by many other papers that are mentioned above.

Lines 386-396: This needs to be revised to be correctly positioned in the major body of literature regarding the importance of HONO vertical structure and the competing roles of ground and aerosol HONO production.

Line 397: The ANN discussion should be presented in a separate section. Given the limited nature of the discussion on the ANN results, this suggests that perhaps the methods section can be reduced and the details of setting up the ANN can be moved to the supporting information.

Line 421: Revise in accordance with new contents of the manuscript after addressing all comments from Reviewers.

Figure 2: Aerosol SA and ground SA need to be considered separately in the ANN.

Figure 3: Is this all hourly data? Not given in the caption. Red and blue traces on every panel is very confusing to interpret. Consider using mainly black lines with solid and different styles of dashing to depict the data or use a wider variety of colors.

Figure 4: Move this to the SI. Add sensitivity test figures to the manuscript.

References

Aubin, D. G. and Abbatt, J. P. D.: Interaction of NO2 with hydrocarbon soot: Focus on HONO yield, surface modification, and mechanism, J. Phys. Chem. A, 111(28), 6263–6273, doi:10.1021/jp068884h, 2007.

Czader, B. H., Rappenglück, B., Percell, P., Byun, D. W., Ngan, F. and Kim, S.: Modeling nitrous acid and its impact on ozone and hydroxyl radical during the Texas Air Quality Study 2006, Atmos. Chem. Phys., 12(15), 6939–6951, doi:10.5194/acp-12-6939-2012, 2012.

Diao, L., Roy, A., Czader, B., Pan, S., Jeon, W., Souri, A. H. and Choi, Y.: Modeling the effect of relative humidity on nitrous acid formation in the Houston area, Atmos. Environ., 131(2), 78–82, doi:10.1016/j.atmosenv.2016.01.053, 2016.

Gonçalves, M., Dabdub, D., Chang, W. L., Jorba, O. and Baldasano, J. M.: Impact of HONO sources on the performance of mesoscale air quality models, Atmos. Environ., 54(2), 168–176, doi:10.1016/j.atmosenv.2012.02.079, 2012.

He, Y., Zhou, X., Hou, J., Gao, H. and Bertman, S. B.: Importance of dew in controlling the air-surface exchange of HONO in rural forested environments, Geophys. Res. Lett., 33(2), 2–5, doi:10.1029/2005GL024348, 2006.

Kim, S., Vandenboer, T. C., Young, C. J., Riedel, T. P., Thornton, J. A., Swarthout, B., Sive, B., Lerner, B., Gilman, J., Warneke, C., Roberts, J. M., Guenther, A., Wagner, N. L., Dubé, W. P., Williams, E. and Brown, S. S.: The primary and recycling sources of OH during the NACHTT-2011 campaign: HONO as an important OH primary source in the wintertime, J. Geophys. Res., 119(11), doi:10.1002/2013JD019784, 2014.

Kleffmann, J., Kurtenbach, R., Lörzer, J., Wiesen, P., Kalthoff, N., Vogel, B. and Vogel, H.: Measured and simulated vertical profiles of nitrous acid - Part I: Field measurements, Atmos. Environ., 37(21), 2949–2955, doi:10.1016/S1352-2310(03)00242-5, 2003.

Liu, Y., Lu, K., Ma, Y., Yang, X., Zhang, W., Wu, Y., Peng, J., Shuai, S., Hu, M. and Zhang, Y.: Direct emission of nitrous acid (HONO) from gasoline cars in China determined by vehicle chassis dynamometer experiments, Atmos. Environ., 169, 89–96, doi:10.1016/j.atmosenv.2017.07.019, 2017.

Maljanen, M., Yli-Pirilä, P., Hytönen, J., Joutsensaari, J. and Martikainen, P. J.: Acidic northern soils as sources of atmospheric nitrous acid (HONO), Soil Biol. Biochem., 67(2), 94–97, doi:10.1016/j.soilbio.2013.08.013, 2013.

Mushinski, R. M., Phillips, R. P., Payne, Z. C., Abney, R. B., Jo, I., Fei, S., Pusede, S. E., White, J. R., Rusch, D. B. and Raff, J. D.: Microbial mechanisms and ecosystem flux estimation for aerobic NO y emissions from deciduous forest soils, Proc. Natl. Acad. Sci. U. S. A., 116(6), 2138–2145, doi:10.1073/pnas.1814632116, 2019.

Oswald, R., Behrendt, T., Ermel, M., Wu, D., Su, H., Cheng, Y., Breuninger, C., Moravek, A., Mougin, E., Delon, C., Loubet, B., Pommerening-Röser, A., Sörgel, M., Pöschl, U., Hoffmann, T., Andreae, M. O., Meixner, F. X. and Trebs, I.: HONO emissions from soil bacteria as a major source of atmospheric reactive nitrogen, Science (80-. )., 341(6151), 1233–1235, doi:10.1126/science.1242266, 2013.

Pinto, J. P., Dibb, J., Lee, B. H., Rappenglück, B., Wood, E. C., Levy, M., Zhang, R. Y., Lefer, B., Ren, X. R., Stutz, J., Tsai, C., Ackermann, L., Golovko, J., Herndon, S. C., Oakes, M., Meng, Q. Y., Munger, J. W., Zahniser, M. and Zheng, J.: Intercomparison of field measurements of nitrous acid (HONO) during the SHARP campaign, J. Geophys. Res., 119(9), 5583–5601, doi:10.1002/2013JD020287, 2014.

Ren, X., Sanders, J. E., Rajendran, A., Weber, R. J., Goldstein, A. H., Pusede, S.

[Figure]

E., Browne, E. C., Min, K. E. and Cohen, R. C.: A relaxed eddy accumulation system for measuring vertical fluxes of nitrous acid, Atmos. Meas. Tech., 4(10), 2093–2103, doi:10.5194/amt-4-2093-2011, 2011.

Rubio, M. A., Lissi, E., Villena, G., Elshorbany, Y. F., Kleffmann, J., Kurtenbach, R. and Wiesen, P.: Simultaneous measurements of formaldehyde and nitrous acid in dews and gas phase in the atmosphere of Santiago, Chile, Atmos. Environ., 43(38), 6106–6109, doi:10.1016/j.atmosenv.2009.09.017, 2009.

Scharko, N. K., Schütte, U. M. E., Berke, A. E., Banina, L., Peel, H. R., Donaldson, M. A., Hemmerich, C., White, J. R. and Raff, J. D.: Combined Flux Chamber and Genomics Approach Links Nitrous Acid Emissions to Ammonia Oxidizing Bacteria and Archaea in Urban and Agricultural Soil, Environ. Sci. Technol., 49(23), 13825–13834, doi:10.1021/acs.est.5b00838, 2015.

Stutz, J., Alicke, B., Ackerman, R., Geyer, A., Wang, S., White, A. B., Williams, E. J., Spicer, C. W. and Fast, J. D.: Relative humidity dependence of HONO chemistry in urban areas, J. Geophys. Res. D Atmos., 109(3), doi:10.1029/2003JD004135, 2004.

Stutz, J., Oh, H. J., Whitlow, S. I., Anderson, C., Dibb, J. E., Flynn, J. H., Rappenglück, B. and Lefer, B.: Simultaneous DOAS and mist-chamber IC measurements of HONO in Houston, TX, Atmos. Environ., 44(33), 4090–4098, doi:10.1016/j.atmosenv.2009.02.003, 2010.

Tong, S., Hou, S., Zhang, Y., Chu, B., Liu, Y., He, H., Zhao, P. and Ge, M.: Exploring the nitrous acid (HONO) formation mechanism in winter Beijing: Direct emissions and heterogeneous production in urban and suburban areas, Faraday Discuss., 189, 213–230, doi:10.1039/c5fd00163c, 2016.

Tsai, C., Spolaor, M., Fedele Colosimo, S., Pikelnaya, O., Cheung, R., Williams, E., Gilman, J. B., Lerner, B. M., Zamora, R. J., Warneke, C., Roberts, J. M., Ahmadov, R., De Gouw, J., Bates, T., Quinn, P. K. and Stutz, J.: Nitrous acid formation in a

snow-free wintertime polluted rural area, Atmos. Chem. Phys., 18(3), 1977–1996, doi:10.5194/acp-18-1977-2018, 2018.

VandenBoer, T. C., Brown, S. S., Murphy, J. G., Keene, W. C., Young, C. J., Pszenny, A. A. P., Kim, S., Warneke, C., De Gouw, J. A., Maben, J. R., Wagner, N. L., Riedel, T. P., Thornton, J. A., Wolfe, D. E., Dubé, W. P., Öztürk, F., Brock, C. A., Grossberg, N., Lefer, B., Lerner, B., Middlebrook, A. M. and Roberts, J. M.: Understanding the role of the ground surface in HONO vertical structure: High resolution vertical profiles during NACHTT-11, J. Geophys. Res. Atmos., 118(17), doi:10.1002/jgrd.50721, 2013.

VandenBoer, T. C., Markovic, M. Z., Sanders, J. E., Ren, X., Pusede, S. E., Browne, E. C., Cohen, R. C., Zhang, L., Thomas, J., Brune, W. H. and Murphy, J. G.: Evidence for a nitrous acid (HONO) reservoir at the ground surface in Bakersfield, CA, during CalNex 2010, J. Geophys. Res. Atmos., 119, 1–14, doi:10.1002/2013JD020971, 2014.

VandenBoer, T. C., Young, C. J., Talukdar, R. K., Markovic, M. Z., Brown, S. S., Roberts, J. M. and Murphy, J. G.: Nocturnal loss and daytime source of nitrous acid through reactive uptake and displacement, Nat. Geosci., 8(1), 55–60, doi:10.1038/ngeo2298, 2015.

Villena, G., Kleffmann, J., Kurtenbach, R., Wiesen, P., Lissi, E., Rubio, M. A., Croxatto, G. and Rappenglück, B.: Vertical gradients of HONO, NO x and O 3 in Santiago de Chile, Atmos. Environ., 45(23), 3867–3873, doi:10.1016/j.atmosenv.2011.01.073, 2011.

Vogel, B., Vogel, H., Kleffmann, J. and Kurtenbach, R.: Measured and simulated vertical profiles of nitrous acid - Part II. Model simulations and indications for a photolytic source, Atmos. Environ., 37(21), 2957–2966, doi:10.1016/S1352-2310(03)00243-7, 2003.

Wang, J., Zhang, X., Guo, J., Wang, Z. and Zhang, M.: Observation of nitrous acid (HONO) in Beijing, China: Seasonal variation, nocturnal formation and daytime budget,

Sci. Total Environ., 587–588, 350–359, doi:10.1016/j.scitotenv.2017.02.159, 2017.

Wojtal, P., Halla, J. D. and McLaren, R.: Pseudo steady states of HONO measured in the nocturnal marine boundary layer: A conceptual model for HONO formation on aqueous surfaces, Atmos. Chem. Phys., 11(7), 3243–3261, doi:10.5194/acp-11-3243-2011, 2011.

Wong, K. W., Lefer, B. L., Rappenglück, B. and Stutz, J.: Vertical profiles of nitrous acid in the nocturnal urban atmosphere of Houston, TX, Atmos. Chem. Phys., 11(8), 3595–3609, doi:10.5194/acp-11-3595-2011, 2011.

Wong, K. W., Tsai, C., Lefer, B., Haman, C., Grossberg, N., Brune, W. H., Ren, X., Luke, W. and Stutz, J.: Daytime HONO vertical gradients during SHARP 2009 in Houston, TX, Atmos. Chem. Phys., 12(2), 635–652, doi:10.5194/acp-12-635-2012, 2012.

Ye, C., Zhou, X., Pu, D., Stutz, J., Festa, J., Spolaor, M., Tsai, C., Cantrell, C., Mauldin, R. L., Campos, T., Weinheimer, A., Hornbrook, R. S., Apel, E. C., Guenther, A., Kaser, L., Yuan, B., Karl, T., Haggerty, J., Hall, S., Ullmann, K., Smith, J. N., Ortega, J. and Knote, C.: Rapid cycling of reactive nitrogen in the marine boundary layer, Nature, 532(7600), 489–491, doi:10.1038/nature17195, 2016.

Young, C. J., Washenfelder, R. A., Roberts, J. M., Mielke, L. H., Osthoff, H. D., Tsai, C., Pikelnaya, O., Stutz, J., Veres, P. R., Cochran, A. K., Vandenboer, T. C., Flynn, J., Grossberg, N., Haman, C. L., Lefer, B., Stark, H., Graus, M., De Gouw, J., Gilman, J. B., Kuster, W. C. and Brown, S. S.: Vertically resolved measurements of nighttime radical reservoirs in Los Angeles and their contribution to the urban radical budget, Environ. Sci. Technol., 46(20), 10965–10973, doi:10.1021/es302206a, 2012.

Zhang, N., Zhou, X., Shepson, P. B., Gao, H., Alaghmand, M. and Stirm, B.: Aircraft measurement of HONO vertical profiles over a forested region, Geophys. Res. Lett., 36(15), doi:10.1029/2009GL038999, 2009.

---

## Author Comment (AC1) · 8 Jun 2020

Response to Review 1

Thank you very much for your constructive comments. The point-by-point responses to your comments are given below. Your comments were all considered, and the manuscript was revised accordingly. In the revised manuscript, all changes are marked in blue.

1. In the INTRODUCTION section: Generally, the introduction section was poorly organized, and it should be rewritten. In this section, many literatures on the HONO

field study were not cited, and the available studies were also not displayed in a line. Especially, the description about tools and methods should be positioned later on the HONO formation mechanism concluded from the field studies during the past years. In a word, this section should be written in more detail. A. The introduction has been thoroughly revised with adequate references being added. In addition, the detailed description about methodology was moved into measurement parts and supplementary information (new). 2. "In this study, we conducted a measurement. . . for two purposed:. . .1 to figure out . . .2 to enhance . . .", However, one can see that which did not coincide with the ABSTRACT structure. Why? A. In this study, we measured HONO, run photochemical and ANN model, and demonstrated its role in O3 production by providing OH radicals in the early morning and its formation through heterogeneous conversion of NO2. These are the main findings of this study and stated in abstract and conclusion, which were also modified in the revised manuscript. 3. In the experimental section, ANN should also be mentioned in the INTRDUCTION sections, as well as the relevant literatures on HONO. A. The detailed description about ANN was moved to supplementary information in the revised manuscript. There, the theoretical backgrounds are explained with relevant literature. 4. In the discussion section, Generally, the discussion about the field data is weak (the measurement period is so short) and it is difficult to support the conclusion. A. Please see section 3.1. It was revised with more details about measurement results. Table 2 and Figure 5 were added and Figure 2, 4, and 6 were modified. In addition, supplementary plots are provided in supplementary information (S3). 5. The English presentation is not so good, which could not be fulfil of the standard of the ACP manuscript. It should be improved greatly before publication. There are many spelling mistakes and syntactic error, as well as unsuitable sentence used. For example, in line 34. ". . .higher in high-O3 episodes (1.82 ppbv) than non-episode (1.20 ppbv)" should be changed to ". . .higher in high-O3 episodes (1.82 ppbv) than that in the non-episode (1.20 ppbv). A. All errors were corrected and the revised manuscript was English proofread as well.

Please also note the supplement to this comment:
https://www.atmos-chem-phys-discuss.net/acp-2019-1012/acp-2019-1012-AC1-
supplement.pdf

---

## Author Comment (AC2) · 8 Jun 2020

Response to Review 2

Thank you very much for your constructive comments. The point-by-point responses to your comments are given below. Your comments were all considered, and the manuscript was revised accordingly. In the revised manuscript, all changes are marked in blue.

Line87 : "The introduction of spectroscopy technique [sic] has facilitated measurement through instrumentation such as Chemical Ionization Mass Spectrometry (CIMS). . .

CIMS is not an optical spectroscopy technique like the others listed. In addition, it is not discussed later in the paragraph when the positives and negatives are described. Yes, CIMS was discussed after spectroscopic techniques. Section 2.2.2 : Section 2.2.2 is difficult to understand. Parameters are weakly defined and technical terminology that is not typical for the atmospheric sciences makes the section challenging to understand for a likely ACP audience (including this reviewer). It is recommended that the authors include a description of how the choices made by the authors impact the outcomes of the ANN approach. We should have been more cautious when introducing new method and terminology. In the revised manuscript, the method section regarding the ANN was shortened and instead, a full explanation is given in supplementary information including the structure of ANN and the detailed procedure of calculation. Lines 263-267 : The average concentrations cited are not necessarily useful numbers because the diel variations in mixing ratios are so large. The nocturnal HONO and daytime O3 mixing ratios are the most important, respectively. Do the 'average' values for the high-O3 and non-episode days reflect the day-long averages? If so, the authors should consider changing the presentation of data to be more applicable to the chemistry at hand Yes, the daily averaged concentration is not useful for O3 in urban areas because of a big difference between the daytime and nighttime. In comparison, the daily average of HONO concentrations reflect the nighttime level well and thus, can be compared from region to region and from time to time. The daily averaged concentrations are to compare the HONO level of Seoul with those of other cities from previous studies. In this study, HONO was discussed in relation with O3 and for this reason, the O3 average concentration was given in pairs with HONO. The various types of concentrations are discussed and compared in the text. For clarity, they are summarized in Table 2 in the revised manuscript. Lines 280-282 : The reason for the anti-correlation of O3 and HONO is due to the source and sink processes for these two gases. While the paper is suggesting that a relationship between these two important reactive trace gases exists, the sources of variation in the diel profile of HONO and O3 are actually *distantly* related. I do not understand why it is appropriate to discuss the data in such a simplistic

way. The inverse relationship between O3 and HONO stems from their inherent pho-
tochemical property. Nonetheless, their daily maximum concentrations were linearly
correlated in this study. These relationships are shown in Figure 5 that was added to
the revised manuscript. The relevant part was modified in the revised manuscript as
follow: In general, HONO and O3 showed an inverse correlation and the overall correla-
tion between the two species was good (r2 = 0.41) (Figure 4). It is mainly derived from
their inherent photochemical property. If the daily maximum concentrations are corre-
lated, O3 concentrations are proportionally increased with HONO (Figure 5). Figures
3 and 4: There is an important discrepancy between the HONO, NO, and NO2 data in
Figs 3 and 4. It appears from inspection of many day-long periods in Figure 3 that NO
and HONO are almost positively correlated. . . the peaks in NO and HONO occur at
very similar times, but the data in Fig 4 show that these peaks occur at different times
(NO is increasing as HONO is decreasing in the morning). Would it be better to plot
the diel mixing ratios after splitting the cases into 'high-O3' and the 'non-episode' peri-
ods? The data, as plotted, are not necessarily consistent. Figure 3 shows time-series
variations of measured species. Of these NOx levels were noticeably high at night in
the beginning of the experiment. It may give an impression of the positive correlation
between NO and HONO. The diurnal variations shown in Figure 4 are pretty typical in
the study region. NO concentrations used to reach the maximum around 7∼8 h in the
morning, followed by NO2 peak. But, NO2 concentrations are higher during the night,
when HONO shows a clear peak. The actual relationship between the species is shown
in Figure RC 1 below, where the correlation with HONO is better for NO2 than NO. The
diurnal variation of HONO and other species are compared for high-O3 episodes and
non-episode in Figure 7. Line 301 : Why does the 80% RH threshold only mean "mist"
vs "haze" in Korea? This seems arbitrary. Perhaps consider the RH dependence of
water uptake to urban aerosol particles (yes, a mixed and perhaps uncharacterized
composition for sure, but there are typical particle types to consider qualitatively). The
Korea Meteorological Administration (KMA) provides the guideline for meteorological
observation, where the 80 % relative humidity is a criterion that distinguishes hydrome-

teors from haze particles. If RH is below 70 % and visibility is between 1 km and 10 km, it is classified as haze condition. All weather phenomena are coded for sharing through WMO (World Meteorological Organization). In the revised manuscript, the relationship between RH and aerosol surface was more clearly stated. The HONO concentration and conversion ratio of $NO_2$ was found to be the highest in the RH range of 70 $\sim$ 80 %, which is the upper and lower margin for haze particles and hydrometeors, respectively. At RH > 80 %, $PM_{2.5}$ concentration was noticeably high (Figure 6). Initially, the Aiken mode (30-120 nm) particles were considered a suitable substrate for heterogeneous conversion of HONO because the dependence of their surface area on RH was similar to HONO concentration and conversion ratio of $NO_2$ (Figure 10). During this experiment, the deliquescence RH of $NH_4NO_3$ and $(NH_4)_2SO_4$ was in the range of 70 $\sim$ 80 %, over which their concentration dramatically increased with the increase in $PM_{2.5}$ mass. When comparing the surface area of Aitken mode (30-120 nm) with condensation mode (120-300 nm) particles according to RH, it increased with RH until 70 % for both two modes. If RH increased over 70 %, however, the behavior of the two modes split: The surface area of Aitken mode particles decreased, but that of condensation mode particles increased, leading to the greatest difference at RH range of 70$\sim$80% (Figure 10). From these results, we concluded that RH played a key role in aerosol transformation, by which processes the availability of aerosol surfaces for the conversion of $NO_2$ was constrained. While Aitken mode particles, which are less susceptible to hygroscopic growth than condensation mode particles, provide surfaces available for HONO formation, HONO concentration was determined by the level of $NO_2$. It was in good agreement with the results of ANN model. This is explained through section 3.3 (Figure 12]. We added this discussion about RH dependence to section 3.3 in the revised manuscript. Figure 5: It may be clearer, or at least less arbitrary, to colorize the markers in Figure 5 by the NO mixing ratio. If a global relationship truly does exist, it will be borne out in the FULL dataset, rather than the arbitrarily chosen >90th percentile of $NO_x$. Figure 5 was modified in the revised manuscript. Color was coded by $NO_2$ concentrations. In this plot, the relationship is not explicit but useful to understand key variables determining HONO concentrations. In Figure RC 3 shown below, the relation between RH and HONO was color-coded with NO, NO2, and NOx. As expected from their diurnal variations, NO is not distinguished in RH-HONO relationship. Figure RC4 and Figure 5 indicate that the level of HONO is dependent on NOx (NO2) concentrations. Line 309 and others: As presented throughout this paper, the HONO mixing ratios are said to be high during high-O3 episodes. Please alter these descriptions to be specific about how HONO is higher *at night* and is associated with elevated daytime ozone. As you suggested, the numbers are summarized and given in Table 2 for quantitative understanding. Eq 10: To what do the t1 and t2 subscripts refer? They are errors and corrected like this: ãĂŰ[OH]ãĂŮ_t1=J_HONO ([HONO]_t1-[HONO]_t2 ), where t1 and t2 are consecutive times. Lines 319-320: "assuming that OH is produced only by HONO". Why do you make this assumption? The modeled diel OH curves shown cannot possibly be dependent only on HONO photolysis – what about the HOx-NOx cycle?? It was corrected. In this study, we tried to calculate the OH produced from HONO photolysis, but not an ambient OH concentrations. Section 3.2 up to Line 338: I think the problem here is in the way that this modeling experiment is presented. Please re-consider how this is presented and clearly indicate the question that you are asking, reasons for simplifying assumptions, and clear statements of conclusions based on this 'toy' experiment with modeling OH from your data. It appears that the authors are trying just to illustrate the impact of HONO on the morning OH mixing ratios, but their descriptions make it seem like their results are broadly accurate, which they are likely not based on their simplifying assumptions. Although it is a simple calculation, the model results demonstrate the role of HONO in O3 formation through HOx cycle. It is already presented from measurement data when they were compared between the high-O3 episodes and non-episode. The difference between the two periods is evident in diurnal variations of O3, HONO, and other precursors (Figure 7). From the model simulation, the contribution of HONO to HOx cycle was quantitatively estimated (Figure 9). This part was revised with presenting all model runs in Figure 8. Lines 336-337: Time integrated values should have some indication of the limits of

the integrals, otherwise these numbers lack real meaning. If the authors simply seek a comparison between the two types of events, then summarizing with a relative % difference may be more appropriate and useful. The integral notation was used to indicate that it was the sum of OH concentrations for several hours of early morning and afternoon, followed by (Alicke et al., 2003). The time interval is given above: early morning (05:00-11:00 LST) and afternoon (12:00-18:00 LST). In the revised manuscript, the relative % difference was added, according to your suggestion. Line 339-346: The sensitivity tests using F0AM say much more than the conclusion that the authors draw. The authors only comment on the importance of HONO to the O3 profile, but the other members of the HOx cycle also play extremely important roles – clearly shown by the model runs!! The authors are encouraged NOT to cherry-pick the results of their studies for the purported benefits of the story that they wish to tell. If the authors would like to highlight the importance of HONO, it would be beneficial to *quantify* the importance of HONO vs other factors. We admit that some statements go further than they mean. For example, VOCs are intimately coupled with HOx cycle, producing O3. When discussing the OH produced from HONO and its impact on O3 production, it was inevitable to state VOC, especially in model simulation. According to your comments, the manuscript was revised with caution. In the abstract, VOC was removed from the last sentence. In original submission, the model results were presented for four scenarios to demonstrate the role of HONO (Figure 7) . In the revised manuscript, the results for all scenarios using BTEX, HCHO, CO, and HONO are presented in Figure 8 and summarized in Table 3. While the contribution of BTEX is the most pronounced, the time for maximum O3 was shifted toward the morning without HONO. These results highlight the importance of HONO in O3 chemistry. Lines 357-364: The role of water in certain HONO production reactions is known. It is recommended that the authors explore RH vs absolute water vapor mixing ratio (or specific humidity). If water participates in the reaction, the kinetics will be related to the absolute concentration of water molecules, rather than a temperature-dependent, saturation-normalized metric like RH. Temperature changes by enough to drive diel variations in RH. This may be

masking the importance of water to the chemistry, or it may be over-emphasizing its role due to coincident diel changes in HONO, NO, NO2, and (temperature-driven) RH. Absolute humidity (AH) was calculated using the following equation, AH (g mˆ(-3))= (6.112×eˆ((17.67×T)/(243.5+T))×RH×2.1674)/(273.15+T). AH was used for calculation of chemical reactions. For comparison, RH was also tested. The results are presented in Figure S3.1, which indicates that RH is more relevant for the formation of HONO than AH. Figure RC 7 (below) compares the relationship of AH and RH with HONO concentrations color-coded by NOx. The overall patterns are similar but, RH shows more consistent relationship with HONO and NOx then AH. These results imply the involvement of heterogeneous reactions in HONO formation mechanism. It was discussed in section 3.3. Lines 380-385: Two items: 1. The response of the conversion ratio to high RH is presented as an independent verification of the behavior of HONO under such conditions. The conversion ratio is essentially a ratio between a measure of [HONO] divided by a measure of [NOx]. If [NOx] is stable over time as it is in this study, then the [HONO]_emission metric is stable over time, and then of course the conversion ratio responds only to [HONO]_formation! The authors have simply shown two ways of expressing the same observation. 2. The connection to Wojtal et al is not very clear, because this is the first time that the marine boundary layer is described at all. Please be more thorough in the description of how the cited work helps to explain your observations. 1) The diurnal variation of NOx is presented in Figure RC 6 below. While the mean concentrations show a typical variation with two peaks in the morning and night, the concentrations vary in a wide range. This diel cycle is also different from HONO variation. 2) The work of (Wojtal et al., 2011) is cited here to emphasize that the dependence of HONO concentrations on RH was observed in previous studies. In addition, they reported the negative relationship between HONO and RH in high RH condition, suggesting the possibility of HONO loss on aerosol surfaces or microlayer of the sea surface in marine boundary layer. Lines 386-396: Multiple comments: 1. Indeed, soot has been discussed as a reactive surface and laboratory experiments have shown that HONO can be formed upon NO2 reaction with soot. However, HONO

can be formed by reaction with many different types of surfaces. Forcing the discussion into black carbon or soot particles reflects a narrow view of the surface catalyzed formation of HONO. In addition, making a broad assumption that all particles between 30-120 nm is FAR oversimplified and arbitrary. Particles in this size range will have a range of compositions, likely with a strong contribution from organic material. Why not simply cut the discussion of black carbon and use the total surface area of particles? 2. PM2.5 is discussed here but the measurement technique and sampling conditions are not described in the paper. 3. Guessing that PM2.5 is measured using a typical type of air quality monitor, PM2.5 could increase due to secondary aqueous phase reactions or simply the growth of particles due to water uptake. Also, particle mass increases as a cube of particle diameter, so PM2.5 measurements may be responding to an entirely different population of particles than the SMPE and MAAP. Also, if the sampling conditions for the SMPS and MAAP measurements are not identical to the PM2.5 measurements, the authors may be ascribing a real effect to a sampling artifact. Please provide sufficient technical information to reassure the reader that this is not the case. Particle measurements require care and nuance, which cannot be assessed by the reviewer or a future reader based on the present manuscript. 1) You are absolutely right regarding the variety of particles available for heterogeneous reactions. Based on the results from previous studies, we examined PM2.5 mass, and Aitken mode (30~120 nm) and condensation mode (120~300 nm) particles as substrates rendering surfaces to HONO heterogeneous reactions. As Aitken mode is similar to the size range of BC (or EC) particles, BC mass was also considered in this analysis. Then, the relation of these variables with RH was examined. Surprisingly, all of them showed a turning point at RH of 70~80% (Figure 10) but in different ways. Of these, the aerosol surface area of the Aitken mode particles showed a similar tendency to NO2 conversion ratio (CHONO). In this study, the RH of 70~80% was found to be critical point affecting the behavior of aerosol composition as well as surface. Sure enough, it was related to the hygroscopic property of inorganic salts controlling PM2.5 mass. It is also the transition RH between dry haze particles from wet hydrometeors, which is defined as meteorological phenomenon by Korea Meteorological Administration. For clarity, this part was completely revised with more information added to the manuscript. Please see the response 6 above. 2) and 3) The detailed measurement techniques and information can be found elsewhere, including (Kim et al., 2020;Jordan et al., 2020) prepared for the KORUS-AQ special issue in Elementa. Lines 397-419: While the authors have used an interesting ANN tool, it is not clear what the results have taught us. We already know that the NOx, surface area, and humidity are key factors that control HONO production. . . we have known this for more than a decade. What can this information provide that will help advance the detailed understanding of HONO formation? We used the ANN model to add quantitative evidence to our findings about heterogeneous HONO formation. At present, there is no firm theory that explains its concentration and formation mechanism, except the conditions constraining HONO concentrations such as high-NOx (urban), high-RH (coastal) or highly particle-polluted sites. With a clue from our measurements about how NOx, surface area, and humidity interplayed in determining HONO concentrations, the ANN model was run. The key contribution of ANN model is what is shown in Figure 12, where the influence of surface area of the Aitken mode particles to HONO concentration was nearly equal to NO2 and greater than RH. Therefore, the results of ANN simulation not only confirm the involvement of heterogeneous conversion of NO2, but also reveals the interplay of the main variables in the process of HONO formation. Consequently, it led to the conclusion of this study that RH constrained the available aerosol surface for HONO formation through intimate coupling with hygroscopic inorganic aerosols and NO2 determined the concentration of HONO as a precursor. Minor overall comments: 1. The use of "%ile" is not typical in formal writing. The authors should consider changing this format from "95 %ile" to "95th percentile". 2. Line 309: "chapter" should be "section" (I stopped making these comments after this point. Substantial English language editing is advised.) They were all changed to percentile. All errors were corrected as far as we could. The revised manuscript was English proofread.

Reference Alicke, B., Geyer, A., Hofzumahaus, A., Holland, F., Konrad, S., Pätz, H.,

Schäfer, J., Stutz, J., Volz‐Thomas, A., and Platt, U.: OH formation by HONO photolysis during the BERLIOZ experiment, Journal of Geophysical Research: Atmospheres, 108, PHO 3-1-PHO 3-17, 2003. Jordan, C., Crawford, J. H., Beyersdorf, A. J., Eck, T. F., Halliday, H. S., Nault, B. A., Chang, L.-S., Park, J., Park, R., Lee, G., Kim, H., Ahn, J.-y., Cho, S., Shin, H. J., Lee, J. H., Jung, J., Kim, D.-S., Lee, M., Lee, T., Whitehill, A., Szykman, J., Schueneman, M. K., Campuzano-Jost, P., Jimenez, J. L., DiGangi, J. P., Diskin, G. S., Anderson, B. E., Moore, R. H., Ziemba, L. D., Fenn, M. A., Hair, J. W., Kuehn, R. E., Holz, R. E., Chen, G., Travis, K., Shook, M., Peterson, D. A., Lamb, K. D., and Schwarz, J. P.: Investigation of factors controlling PM2.5 variability across the South Korean Peninsula during KORUS-AQ, Elementa: Science of the Anthropocene, in review, 2020. Kim, H., Gil, J., Lee, M., Jung, J., Whitehill, A., Szykman, J., Lee, G., Kim, D., Cho, S., Ahn, J., Hong, J., and Park, M.: Overview and characteristics of air quality in the Seoul Metropolitan Area during the KORUS-AQ campaign, Elementa: Science of the Anthropocene, in review, 2020. Wojtal, P., Halla, J., and McLaren, R.: Pseudo steady states of HONO measured in the nocturnal marine boundary layer: a conceptual model for HONO formation on aqueous surfaces, Atmospheric Chemistry and Physics, 11, 3243-3261, 2011.

Please also note the supplement to this comment:
https://www.atmos-chem-phys-discuss.net/acp-2019-1012/acp-2019-1012-AC2-supplement.pdf

—————————————————————

---

## Author Comment (AC3) · 8 Jun 2020

Referee Comment 3

Thank you very much for your constructive comments. The point-by-point responses to your comments are given below. Your comments were all considered, and the manuscript was revised accordingly. In the revised manuscript, all changes are marked in blue.

<Major comments> 1. Nocturnal HONO formation chemical mechanisms are not correct. There is clear confusion in the presentation of what HONO production processes

are relevant to exploring its nocturnal formation. The literature has not been sufficiently reviewed and puts the results of this study into question. In the introduction, the Authors present reactions 2 through 4 which have never been demonstrated in many prior reports to be important under nocturnal conditions. Further to this, reaction 4 is for the photolysis of ortho-nitrophenols, not from reaction of NO2. The Authors seem confused on this mechanism and it generates much skepticism regarding the quality of their model results if established HONO chemistry is presented erroneously here. Since release of HONO from ortho- and para-nitrophenols requires light, it is also invalid to explore for HONO production chemistry at night. This reaction has not been shown to be important in polluted daytime atmospheres for HONO production. How could it possible be important to nocturnal chemistry? Given the focus of this manuscript on nocturnal HONO production, the Authors need to revisit the state of knowledge for relevant nocturnal reactions to consider (e.g. has NO2 + NO + H2O ever made a significant amount of HONO in any prior report?) and rerun their ANN from a relevant starting point. Lines 119-123 in the Introduction are irrelevant to this work as this is all daytime HONO chemistry. Direct emissions from soils have not been quantified for diurnal roles (Lines 127-129) so are of limited utility. Seminal works for such investigations are also not cited correctly, a small selection includes: (Maljanen et al., 2013; Mushinski et al., 2019; Oswald et al., 2013; Scharko et al., 2015). The Authors then proceed to describe heterogenous reactions as being universally catalytic, which simply is not true. Again, chemical mechanisms relevant to the day time (photoexcited humic substances) are conflated with marginally relevant nocturnal sources (fresh soot emissions (Aubin and Abbatt, 2007)) raising major questions as to how the quality of the subsequent data analysis can be trusted. Further to this, there is a major oversight on at least 15 years of research reporting on the role of ground and aerosol surfaces in producing nocturnal HONO and the resultant vertical structure of HONO in the nocturnal atmosphere. A small selection of this body of research includes: (Kleffmann et al., 2003; Stutz et al., 2004; VandenBoer et al., 2013; Villena et al., 2011; Wong et al., 2011; Young et al., 2012). Nocturnal HONO production, in short, is a complex function of chemistry and mixing, often resulting in a complex vertical structure, which has shown in many polluted environments to be highly linked to ground surface conversion of NO2 with a water dependence. The Authors need to be clear in their understanding of the relevant chemical and physical processes governing nocturnal HONO production prior to exploring which is most important to O3 production via the ANN. Linking this understanding to the set up of the ANN is critical to validating the approach used in Section 2.2.2, for example. As this currently stands, this particular section is not possible to evaluate with a sound basis in HONO nocturnal chemistry and requires revision. First of all, the reaction 4 was corrected. It is right that some reactions on the list are not important at nighttime. Especially, the reactions 3~5 were from theoretical and lab experiments but used in modelling studies. The reactions 2~9 indicate all possibilities of HONO formation including daytime as well as nighttime. The main objectives of this study are to demonstrate the impact of HONO on O3 formation and to understand the HONO formation mechanism. To achieve these goals, we used two different models. Unfortunately, the effect of ground surface was not incorporated into the ANN model because there was no measurement to represent its effect. The loss due to deposition to the ground surface cannot be ignored, when being estimated based on previous studies. Since the recent studies suggest various sources of HONO even during the day, the source of HONO from ground surface will have to be investigated in a separate study with a different approach such as chamber or vertical structure measurement. Upon your suggestion, we delved into the aerosol surface areas and refined the result of ANN model, which demonstrates the feature importance of physicochemical factors to HONO formation. As a results, the manuscript was thoroughly revised. 2. Given the focus of this work on the use of ANN to model nocturnal HONO chemistry, the Introduction and results should do a much better review and contrast to the existing capabilities of other models such as CMAQ (Czader et al., 2012; Diao et al., 2016; Gonçalves et al., 2012) or custom models (Kleffmann et al., 2003; Tsai et al., 2018; Vogel et al., 2003; Wong et al., 2012). The Authors should review this literature to aid in preparations for the re-running of their ANN. A. We really appreciated your detailed information and suggestion. We added these references to revised manuscript. 3. The quality control for the HONO dataset, as presented in the methods section (and is entirely absent from the results/discussion) is not thorough. The Authors state that optical measurements are free of sampling artifacts, yet all instruments have inlets with surfaces where NO2 can react with water to produce HONO, a phenomenon described for most field instruments along with an applied correction for such an 'inlet effect'. The best studies also account for non-linearities associated with changing NO2 and water concentrations. No sampling information for the HONO measurements is given. Were inlets used? How long were they? What were they made out of? What was the sample flow through the inlet? Was any particulate matter exclusion done on the sample flow? Was the NO2 inlet effect in producing HONO quantified and corrected for? The intercomparison made here is also weak. An r-value of 0.74-0.84 is very bad and a correlation alone is insufficient to describe the comparability of the measurements. Slope, intercept, and their associated uncertainties are also necessary to calculate as part of measurement validation. These plots should be provided in a supporting information document. The Authors then go on to state that they made 1 Hz measurements of HONO and averaged these to 1 hour time points. Why was this done? Lines 168-172 suggest that the instrument was not operated correctly to make very good high time resolution measurements, yet no quality control analysis of the data is presented. How can one trust this HONO measurement? Justification for the approach used to arrive at a reliable final HONO dataset must be explained. Further to this, the Author's state a theoretical detection limit, but do not state what timescale this applies to. Presumably if they are averaging 1 Hz data to an hourly timescale, the detection limit is much better than 0.1 ppb? Or was the instrument operated so poorly that they were worse? Were field blanks performed on the instrument by overflowing the inlet with zero air? If yes, these should have been used to calculate the instrument detection limits that existed for this campaign, not the theoretical detection limits. Please perform these analyses to demonstrate that the HONO measurements are reliable and of high quality (or at least possible to clearly understand the potential limitations in their quality). As it currently

stands, the Authors discuss data below the theoretical detection limit in Section 3.1 and one wonders how or why they would report this? The method section was completely rewritten because it focused on the measurement of QC-TILDAS.

<Detail comments>

Line 30: 'but the. . .. Unclear'. This is not part of the manuscript objective. Remove from here. It was removed from the abstract. Lines 32-46: Nearly the entire abstract has disjointed writing that is very difficult to follow the logic of. Revise according to major changes to findings/conclusions based on major revisions, but also for clarity. In particular, the sentence from Lines 39-41 makes no sense to me, even with a thorough understanding of HONO atmospheric chemistry. Also, at Lines 42-43 'surface area' is listed but what it applies to (i.e. aerosols or ground or both) is not clear. Thank you for your advice. It was changed to 'aerosol surface area' for clarity. The abstract was revised, too. Line 60: 'huge' is unnecessary to use here. Sensational modifiers throughout the manuscript should similarly be removed. The data will communicate the exact value and direction of changes, so readers can evaluate this objectively. It was modified as follows: This implies that HONO strongly affects the early morning photochemistry cycle, along with the promotion of VOC oxidation, causing high O3 concentrations in afternoon. Line 80: 'lowing' should be 'lowering' This part was removed in the revised manuscript. Lines 83-86: If you are going to mention real-time IC techniques for HONO, there are earlier studies with better analytical results from the group of Jennifer Murphy (U Toronto) (VandenBoer et al., 2014). There is also a substantial amount of work from Jack Dibb (U New Hampshire) collecting HONO by mist chamber IC that have demonstrated good analytical capabilities in intercomparisons with DOAS (Pinto et al., 2014; Stutz et al., 2010). Also missing from this section is discussion of a substantial body of measurements made by broadband cavity enhanced absorption spectrometry (BBCEAS). Finally, given the lack of discussion on the performance and quality of the QC-TDLAS HONO measurements in this study, one is left wondering what the purpose of summarizing all of these analytical techniques is? Your helpful comments are very

much appreciated. Following the other reviewer's comment, however, the description of measurement technique was shortened in Introduction. Instead, the details about QC-TIDAS measurement was given in section 2.1. Measurement. Lines 95-96: The Authors have the sensitivity of the LOPAP backwards. It is very good a detecting 'low mixing ratios of HONO'. The detection limits of the LOPAP are often below 1 pptv if integrated over 5 minutes or more. Furthermore, there are many custom built stripping coil instrument references that are missing here (Ren et al., 2011; Ye et al., 2016; Zhang et al., 2009). This information was added to the introduction. Line 102: 'TIDLAS' should be 'TDLAS It was corrected. Line 103: 'albeit' should be 'despite' It was corrected. Lines 152-154: The TDLAS technique has been described three times up to this point in the manuscript and denoted with three different descriptions and abbreviations. Pick one and use it throughout. It was corrected. Line 187: Reference formatting is incorrect and needs to be revised throughout. It was corrected. Lines 195-197: Why is the dilution factor the only parameter adjusted in the model that is worth mentioning in the methods? This sensitivity test comes out of nowhere and no justification is provided for why this is a sound approach. In the revised manuscript, this part was modified as follows: As a first-order dilution rate constant in the model, the dilution factor (kdil) was adjusted to reduce uncertainties due to the lack of explicit representation of transport processes. As a variable of the model, the dilution factor needs to be adjusted before it was run. It is the inherent property of the F0AM model that we used in this study. The dilution factor is analogous to physical loss lifetime and was introduced to represent transport processes. The result of sensitivity test is given in section 3.2 and shown in Figure 8. Line 197: 'session 3.3' should be 'Section 3.3' It was corrected. Lines 199 – 201: The Authors define Artificial Neural Network (ANN) over and over. Do it once at the first mention and simply use ANN moving forward. The second instance of its use at Line 201 has been written incorrectly. The most part of section 2.3.2 regarding ANN was moved into Supplementary Information. Details of the ANN model and its running procedure is provided there. Lines 205-206: The use of ANN for quantum states of HONO is irrelevant to this work. Remove. It was deleted. Line 208: Revise to '. . .of

each neural network node with a clear chemical or physical process It was modified as you recommended. Line 211: Check citations for correctness of Author names and all other details according to ACP guidelines for Authors Yes, they are correct. Lines 216-218: But what are you doing, and why, with the calculation in Equation 1? This is very unclear. Revise As input variables, the measurement data are in wide range. For example, HONO in 0 ∼ 4 ppbv and MLH in 0 ∼ 3000 m. If these values are not normalized and used in the model, the result will be more dependent on large values, thereby being biased. Therefore, all variables were normalized to have equal weight using (Eq S2.2). Line 260: '%ile' is incorrect. This is 'percentile'. Fix throughout. All of them were corrected as percentile. Line 264: 'average O3 and HONO' What kind of average is this? 24-hour average? Clarify. The presented data does not look like it has any 24-hr period with more than 1 ppbv of HONO. Line 268: 'High' should be 'high' There are various measurement statistics stated in the text. For clarity, they are summarized in Table 2 in the revised manuscript. For example, the average concentrations of HONO and O3 are given for daytime and nighttime during the high-O3 episode and non-episode. For the two episodes, the concentrations are the average for episode days. High was changed to high. Line 277: The dataset for HONO would be expected to be log-normal distributed. Here and above, the authors are discussing the 95th percentile with an implicit meaning of significance, yet if the dataset is not log-transformed, then the 95th percentile would be a biased value to report due to the skewed distribution from a normal distribution. As you said, HONO concentrations show lognormal distribution. The top 10 percentile concentrations are generally used to represent the high concentrations of the measurement. In case of O3, the 90th percentile is considered relevant for comparison in background sites for a long time. In comparison, the 99.9th percentile of 1 hr-O3 represents the maximum concentration in urban areas. This study was conducted as an intensive experiment for a relatively short period, about a month. In this case, the 95th percentile would be appropriate for representing the maximum concentration. Lines 291-294: This is suggestive of strong direct emissions. See many recent papers from Beijing and perform the same analysis for Seoul (Liu et al., 2017; Tong et

al., 2016; Wang et al., 2017) instead of choosing an estimate for this work. The recent studies were cited in the revised manuscript. In this study, we gave it a try to understand the HONO formation using the ANN model. For this purpose, the estimation of direct emission of HONO was necessary, which is described in [Line 325] of the revised manuscript. Line 304: The Authors conclude here that aerosol surfaces are important, but their lack of awareness of literature reports on the role of the ground surface relative to aerosol surfaces being far more important suggests that they need to reconsider their interpretation of the data after more thoroughly reviewing the literature (see references given in Major Comments above). All vertically-resolved HONO measurements have shown that the ground is a major sink for HONO, as well as dew and available surface water (He et al., 2006; Rubio et al., 2009; Stutz et al., 2004; Tsai et al., 2018; VandenBoer et al., 2015; Wojtal et al., 2011). Based on the study of (VandenBoer et al., 2013), the surface loss of HONO is expressed as loss-1/4 $\gamma\_(HONO,ground)$ RH/20 C_HONO ãĂŰ[HONO]ãĂŮ_ground. This equation yields 0.007 $\sim$ 0.041 with the average of 0.025 ($\gamma\_(HONO,ground)$=8.7×ãĂŰ10ãĂŮ^(-5)), and 0.005 $\sim$ 0.039 with the average of 0.022 ($\gamma\_(HONO,ground)$=7.3±1×ãĂŰ10ãĂŮ^(-5)) for this study. The HONO measurement was conducted under 10 m heights, therefore observed ambient HONO can represent [HONO]ground (VandenBoer et al., 2013). This result indicates that the surface loss is about 0.5 $\sim$ 4.1% of the ambient HONO concentration. This ground sink is not negligible, but was not considered in this study. In addition to the assumption and simplification stated above, it does not affect the main results of this study and is within the uncertainty of model simulation. Line 319-320: How can the Authors justify ignoring the photolysis of O3 and aldehydes as major sources of OH during KORUS-AQ. This does not seem reasonable. Even in winter atmospheres, this additional chemistry leading to OH production has been shown to be critical to capture (Kim et al., 2014). We admit that it is a wrong expression and corrected in the revised manuscript. Here, we did not estimate the ambient OH concentration, but the OH produced from HONO photolysis. In the early morning around 5 am, when O3 concentration is at its minimum level (< 10 ppbv), it is reasonable to assume that there is no source for OH other

than HONO photolysis. Thus, we calculated the OH produced from HONO photolysis, which was compared for the high-O3 episodes and non-episode. Lines 336-337: Why is this integral notation being used? The discrete time intervals being used are not given and the explanation here is very unclear. Revise. The integral notation was used to indicate that it was the sum of OH concentrations for several hours of early morning and afternoon, followed by (Alicke et al., 2003), but not an hourly concentration. The time interval is given above: early morning (05:00-11:00 LST) and afternoon (12:00-18:00 LST). In the revised manuscript, the relative % difference was added for clarity. Lines 341-346: Add line colors to the S# references here so they do not get confused with figures in the supplemental information. Or change the identification codes for the model runs to not start with 'S'. Thank you for your insightful suggestion because the supplementary information was added to the revised manuscript. Model scenarios are identified with M instead of S. Lines 357-359: 'physical mechanisms as well as'. This is nonsense. Heterogeneous hydrolysis is a chemical mechanism. Remove. Yes. It was changed to "heterogeneous chemical reactions" in the revised manuscript Line 360: Exploring the role of NO (and R7), based on all known nocturnal HONO chemistry is useless and should be removed throughout the manuscript. There is no basis for this from the literature. In this experiment, NO concentration showed a big change from period to period (Figure 3). When the aged air mass was encountered (May 24~26), NO came down below the detection limit. In contrast, it was highly elevated particularly at night under stagnant condition (May 17~22). This diurnal pattern of concentration is presented in Figure 4. In recent studies conducted at high NOx environment, NO was found to contribute to HONO formation (Tong et al., 2015;Zhang et al., 2019). In the ANN model of this study, it turned out that the contribution of NO was the largest among input 8 variables. Thus, it was explained as follows: The largest contribution of NO was primarily due to the chemical coupling with NO2, but partly reflects the daytime formation of HONO, albeit its low concentrations. You are absolutely right that R7 is slow and negligible in terms of HONO formation. However, the formation mechanism of HONO has not unequivocally understood, yet. Thus, R7

had better be in the list of reactions, just as a possibility. Line 366: 'considered HONO photolysis, SZA, and MLH' This is nighttime data. Why is photochemistry being considered? The reasoning here is almost impossible to understand. The use of SZA was not to consider photochemical reactions, but to select the times (0∼5 am LST) during which HONO is not photolyzed. The nighttime was determined as 7 pm ∼ 5 am when SZA was over 90 degrees. Lines 371-373: It is the conversion of NO2 that is typically used for this calculation, not NOx, which is consistent with the hydrolysis of NO2 being the dominant chemical source of HONO to the nocturnal atmosphere. The Authors should use NO2 conversion efficiency in this work. Given the discussion that follows on Lines 377-380 about NO2 conversion, perhaps this is a typo? We agree with you and it was done with NO2 instead of NOx. Line 381: Figure 9a. Fix notation. It was changed as CHONO,NO2 Lines 382-385: The work cited here made measurements over open marine water. The dataset in this work is collected over an urban landscape. The uptake of NO2 to ground surfaces has been described in detail by many other papers that are mentioned above. This reference was cited here to emphasize that the tendency of HONO depending on RH was observed in previous studies, but not to compare the HONO concentrations. Lines 386-396: This needs to be revised to be correctly positioned in the major body of literature regarding the importance of HONO vertical structure and the competing roles of ground and aerosol HONO production. This part was completely revised. The HONO concentration and conversion ratio of NO2 was found to be the highest in the RH range of 70 ∼ 80 %, which is the upper and lower margin for haze particles and hydrometeors, respectively (Figure 10). At RH > 80 %, PM2.5 concentration was noticeably high. Initially, the Aiken mode (30-120 nm) particles were considered a suitable substrate for heterogeneous conversion of HONO because the dependence of their surface area on RH was similar to HONO concentration and conversion ratio of NO2. During this experiment, the deliquescence RH of NH4NO3 and (NH4)2SO4 was in the range of 60 ∼ 80 %, over which their concentration dramatically increased with the increase in PM2.5 mass. When comparing the surface area of Aitken mode (30-120 nm) with condensation mode (120-300 nm)

particles according to RH, it increased with RH until 70 % for both two modes. If RH increased over 70 %, however, the behavior of the two modes split: Aitken mode particles decreased, but condensation mode particles increased, leading to the greatest difference at RH range of 70~80%. From these results, we concluded that RH played a key role in aerosol transformation, by which processes the availability of aerosol surfaces for the conversion of NO2 was constrained. While Aitken mode particles, which are less susceptible to hygroscopic growth than condensation mode particles, provide surfaces available for HONO formation, HONO concentration was determined by the level of NO2. It was in good agreement with the results of ANN model. This is explained through section 3.3 (Figure 12). Line 397: The ANN discussion should be presented in a separate section. Given the limited nature of the discussion on the ANN results, this suggests that perhaps the methods section can be reduced and the details of setting up the ANN can be moved to the supporting information. We provide the detailed information and methodology about ANN in supplementary information. Line 421: Revise in accordance with new contents of the manuscript after addressing all comments from Reviewers. The conclusion was rewritten. Figure 2: Aerosol SA and ground SA need to be considered separately in the ANN. It was changed to aerosol surface area (ASA). Figure 3: Is this all hourly data? Not given in the caption. Red and blue traces on every panel is very confusing to interpret. Consider using mainly black lines with solid and different styles of dashing to depict the data or use a wider variety of colors. All data are hourly averaged. This figure looks busy, no matter what colors are used. The red and blue colors are used to distinguish the two plots in one panel. We found that this way of presentation was better than 12 panels of single species. If it is not necessary to show the time series variations of chemical species in the manuscript, it can be moved to supplementary information. Figure 4: Move this to the SI. Add sensitivity test figures to the manuscript. Figures were changed in the revised manuscript

Reference Alicke, B., Geyer, A., Hofzumahaus, A., Holland, F., Konrad, S., Pätz, H., Schäfer, J., Stutz, J., Volz‐Thomas, A., and Platt, U.: OH formation by HONO photolysis during the BERLIOZ experiment, Journal of Geophysical Research: At-

mospheres, 108, PHO 3-1-PHO 3-17, 2003. Tong, S., Hou, S., Zhang, Y., Chu, B., Liu, Y., He, H., Zhao, P., and Ge, M.: Comparisons of measured nitrous acid (HONO) concentrations in a pollution period at urban and suburban Beijing, in autumn of 2014, Science China Chemistry, 58, 1393-1402, 2015. VandenBoer, T. C., Brown, S. S., Murphy, J. G., Keene, W. C., Young, C. J., Pszenny, A., Kim, S., Warneke, C., de Gouw, J. A., and Maben, J. R.: Understanding the role of the ground surface in HONO vertical structure: High resolution vertical profiles during NACHTT‐11, Journal of Geophysical Research: Atmospheres, 118, 10,155-110,171, 2013. Zhang, W., Tong, S., Ge, M., An, J., Shi, Z., Hou, S., Xia, K., Qu, Y., Zhang, H., and Chu, B.: Variations and sources of nitrous acid (HONO) during a severe pollution episode in Beijing in winter 2016, Science of the Total Environment, 648, 253-262, 2019.

Please also note the supplement to this comment:
https://www.atmos-chem-phys-discuss.net/acp-2019-1012/acp-2019-1012-AC3-supplement.pdf